# Non-Invasive Cardiac Output Measurement Using Inert Gas Rebreathing Method during Cardiopulmonary Exercise Testing—A Systematic Review

**DOI:** 10.3390/jcm12227154

**Published:** 2023-11-17

**Authors:** Agnieszka Chwiedź, Łukasz Minarowski, Robert M. Mróz, Hady Razak Hady

**Affiliations:** 1I Department of General and Endocrine Surgery, Medical University of Bialystok, 15-276 Bialystok, Poland; 2II Department of Lung Diseases and Tuberculosis, Medical University of Bialystok, 15-540 Bialystok, Poland

**Keywords:** aerobic performance, cardiopulmonary exercise test, cardiac output, non-invasive measurement of cardiac output, inert gas rebreathing

## Abstract

Background: The use of inert gas rebreathing for the non-invasive cardiac output measurement has produced measurements comparable to those obtained by various other methods. However, there are no guidelines for the inert gas rebreathing method during a cardiopulmonary exercise test (CPET). In addition, there is also a lack of specific standards for assessing the non-invasive measurement of cardiac output during CPET, both for healthy patients and those suffering from diseases and conditions. Aim: This systematic review aims to describe the use of IGR for a non-invasive assessment of cardiac output during cardiopulmonary exercise testing and, based on the information extracted, to identify a proposed CPET report that includes an assessment of the cardiac output using the IGR method. Methods: This systematic review was conducted by PRISMA (Preferred Reporting Items for Systematic Reviews and Meta Analyses) guidelines. PubMed, Web of Science, Scopus, and Cochrane Library databases were searched from inception until 29 December 2022. The primary search returned 261 articles, of which 47 studies met the inclusion criteria for this review. Results and Conclusions: This systematic review provides a comprehensive description of protocols, indications, technical details, and proposed reporting standards for a non-invasive cardiac output assessment using IGR during CPET. It highlights the need for standardized approaches to CPET and identifies gaps in the literature. The review critically analyzes the strengths and limitations of the studies included and offers recommendations for future research by proposing a combined report from CPET-IGR along with its clinical application.

## 1. Introduction

Cardiopulmonary exercise testing (CPET) offers the investigator the unique opportunity to study cellular, cardiovascular, and ventilatory system responses under controlled conditions of metabolic stress [1].

Cardiopulmonary exercise testing (CPET) is a versatile and comprehensive diagnostic tool that has gained popularity and is now utilized across various demographics and patient groups [2]. Previously, CPET was primarily performed on healthy individuals or those with specific cardiovascular conditions. However, its application has broadened significantly, and it is now employed in diverse populations. CPET is routinely conducted on both men and women, allowing for a gender-specific assessment of exercise capacity and functional limitations. It provides valuable insights into the physiological responses of different genders during exercise and helps identify any gender-specific exercise limitations [3]. 

Furthermore, CPET is not limited to specific age groups; it is now employed for children [4,5] and the elderly [6]. This allows for the evaluation of exercise performance and the identification of any age-related limitations.

In addition to studying healthy individuals and those with specific conditions, CPET is now applied to assess exercise capacity in both athletic and frail populations [7,8]. By comparing the performance of highly trained athletes with individuals who are frail or have reduced functional capacity, CPET helps determine the impact of physical fitness on exercise performance and identifies the limitations faced by the frail population during physical activity. 

CPET has evolved beyond examining exercise performance and peak oxygen uptake alone. It now aims to answer broader questions about the underlying causes of exercise limitations. By analyzing various parameters during CPET, such as heart rate, blood pressure, gas exchange, and ventilatory responses, medical professionals can identify which organ or bodily function is responsible for the exercise limitation [9,10]. This information aids in the diagnosis and treatment of various conditions impacting exercise capacity. Moreover, CPET plays a crucial role in determining the prognosis of patients, being the strong and independent marker of risk for cardiovascular and all-cause mortality [11,12,13,14,15]. By assessing exercise performance and physiological responses, healthcare providers can predict a patient’s condition’s potential outcome, helping to make informed decisions regarding their treatment plans and interventions. What is more, CPET is considered an ideal pre-operative assessment tool, as it provides an objective measure of fitness or functional capacity [16]. It assesses the comprehensive function of the cardiac, circulatory, respiratory, and muscle metabolic systems during physiological stress. Further, it can pinpoint the underlying cause of exercise intolerance. The anaerobic threshold (AT) measured through CPET testing strongly predicts mortality resulting from cardiopulmonary causes during the postoperative period. By conducting preoperative screening with CPET testing, high-risk patients can be identified, enabling the selection of appropriate perioperative management strategies [17]. The current guidelines from the American Heart Association, American College of Cardiology [18], and the European Society of Cardiology [19] suggest that assessing exercise capacity, when feasible, can provide valuable insights in the evaluation of noncardiac surgery for high-risk patients with an unknown functional capacity.

Nevertheless, the European Society of Cardiology stated, “The full potential of CPET in the clinical and research setting remains underused” [10]. Despite its intrinsic advantages, CPET also has limitations due to how it assesses cardiac and pulmonary function, i.e., in the case of the heart, it only provides an indirect assessment of cardiac function as a pump. In contrast, in the case of the lungs, it gives intermediate AT values unless the method is extended to include arterial blood gasometry during the test. This review considers the use of the already-known IGR (Inert Gas Rebreathing) method to assess the direct cardiac function during CPET. This is presented in detail in this paper.

The inert gas rebreathing method is a non-invasive technique used to assess cardiac output (CO) and pulmonary blood flow (PBF) during exercise. It involves the rebreathing of a gas mixture containing inert gases, which are soluble and enter the bloodstream via pulmonary diffusion without binding to hemoglobin [20]. 

This method allows for the measurement of respiratory gas exchange and airflow, providing additional information on oxygen utilization and ventilation [21].

The inert gas rebreathing method has been shown to be a safe and reliable alternative to invasive and expensive techniques for measuring CO, such as thermodilution [22]. It has been used in numerous studies to assess cardiac function in patients with heart failure [22] and to evaluate the accuracy and precision of different rebreathing techniques [20,22,23]. The results of these studies have demonstrated that the inert gas rebreathing method can provide reasonable estimates of CO and pulmonary blood flow, with some variations in accuracy and precision depending on the specific gases used and the exercise intensity [20,23].

The use of CPET and IGR provides us with an even broader understanding of the clinical state of the patient under examination through lots of symptom-limited, incremental exercise, commonly in combination with the comprehensive breath-by-breath monitoring of cardiopulmonary variables (e.g., peak oxygen consumption/maximal oxygen consumption (peak VO_2_/VO_2_ max), pulmonary CO_2_ output (VCO_2_), minute ventilation (VE), heart rate (HR), cardiac output (CO), stroke volume (SV), and subjective responses (e.g., dyspnea and leg pain)) and, as needed, measurements such as exercise-related arterial oxygen desaturation and dynamic hyperinflation [3]. 

### 1.1. Cardiac Output Measurement

It is useful to add the measurement of CO, which is considered the best indicator of cardiac function during exercise, to the values described above [24]. The measurement of CO represents an added value to a standard CPET.

CO is defined as the volume of blood pumped by the heart per minute. It is an important parameter that reflects the ability of the heart to supply blood to the body’s tissues. CO is calculated as the product of the heart rate (HR) and stroke volume (SV), which is the volume of blood pumped by the heart with each beat.

CO can be expressed mathematically as CO = HR × SV. CO is a crucial determinant of oxygen delivery to the body’s tissues, and it can be affected by several factors, including heart rate, contractility, preload, and afterload [25]. Measuring CO is important for diagnosing and managing heart and circulatory conditions [26,27,28] and monitoring the response to therapy [29,30,31]. 

There are several methods of CO measurements. CO can be measured during pulmonary artery catheterization (PAC) using the thermodilution method described by W. T. McGee et al. [32]. The thermodilution method provides a relatively accurate and continuous CO measurement [33], and it is widely used in critical care settings. However, it is an invasive procedure and carries a risk of complications [34], including infection [35], bleeding [36], and arrhythmias [37,38]. Non-invasive methods for measuring CO include bioimpedance [39], bioreactance [40], pulse contour analysis [41], pulse wave velocity [42], photoplethysmography [43], partial CO_2_ rebreathing [44], and, finally, inert gas rebreathing (IGR) [45].

### 1.2. Inert Gas Rebreathing

IGR is a non-invasive technique used to measure CO and PBF. The use of IGR has been shown to produce pulmonary blood flow measurements that are comparable to those obtained by various other methods such as thermodilution, direct Fick, and modified Fick methods during rest as well as graded exercise [46,47,48,49]. Furthermore, this method offers high reproducibility for CO measurements during rest and exercise for both healthy individuals and those with cardiac indications [50,51]. 

The IGR method may use different neutral gases, i.e., nitrogen, helium, acetylene, carbon dioxide, and sulfur hexafluoride [52]. Nitrogen is an inert gas that is relatively insoluble in blood and tissue, making it an ideal tracer for measuring blood flow. Xenon gas is also used in inert gas rebreathing, as it has low solubility in blood and a high tissue-to-blood partition coefficient, which makes it a useful tracer for measuring pulmonary blood flow [53]. Due to its low solubility in blood, helium is often used as a diluent in the rebreathing mixture to increase the tracer gas concentration and improve measurement accuracy [53].

Acetylene is an inert gas used as a tracer gas in some IGR methods. However, compared to other commonly used tracer gases, acetylene is more soluble in blood and tissue and, therefore, is less ideal for measuring blood flow [54].

Another commonly used gas is carbon dioxide, as it is readily metabolized by the body and can provide information about gas exchange in the lungs [55].

Finally, sulfur hexafluoride (SF_6_) is also used as a tracer gas in IGR, which has a high molecular weight and is even less soluble in blood than nitrogen, making it an effective tracer for measuring CO [56].

### 1.3. IGR with the Use of SF_6_

The IGR technique involves a machine inflating a bag with an oxygen-enriched mixture of an inert soluble gas (0.5% nitrous oxide) and an inert insoluble gas (0.1% sulfur hexafluoride—SF_6_), which is an inert gas that does not alter the body’s metabolism [45,57,58,59,60,61]. Patients breathe into a respiratory valve via a mouthpiece and a bacterial filter with a nose clip, and at the end of expiration, the valve automatically activates so that patients rebreathe from the prefilled bag for 10 to 20 s. The CO measurement is taken by a photoacoustic analyzer over a three to five breath interval. The lung volume is determined by sulfur hexafluoride, while the nitrous oxide concentration decreases during rebreathing with a rate proportional to the PBF [45,57]. CO equals the PBF only if SpO_2_ is above 98% on the pulse oximeter, indicating no pulmonary shunt flow. If SpO_2_ is below 98%, CO equals the PBF plus the shunt flow [57].

This method can be successfully performed both at rest and during CPET [45,51]. 

A standard IGR report typically includes resting measurements such as CO, CI (Cardiac Index), VO_2_/kg (Oxygen consumption per kilogram of body weight), VO_2_/min (Absolute oxygen consumption per minute), SV (Stroke Volume), (a-v)O_2_ difference (Oxygen content difference between arterial and venous blood), and shunt (%). The report may also feature graphs showing the normalization curves of the gases used for measurement. 

If the IGR test is performed during an exercise protocol, the report includes additional results from the breath-by-breath exercise assessment. These results may consist of a maximum load (highest workload achieved), maximum heart rate, respiratory rate, VCO_2_ (Ventilatory Carbon Dioxide Output), VO_2_ (Oxygen Consumption), RQ (Respiratory Quotient), VE/VCO_2_ (Ratio of minute ventilation to carbon dioxide output), saturation levels, and other typical outcomes.

The process of performing an IGR during CPET is presented in Figure 1.

The contents of the standard IGR report are shown in Figure 2.

The SF_6_ concentration measurement is repeated several times during the exercise challenge and the CO is calculated using a specialized computer software (Innocor™, Innovision, Odense, Denmark, ver. 8.10).

CO can be derived from the change in the SF_6_ concentration in the exhaled air, which reflects the amount of SF_6_ that has diffused into the blood and been transferred to the lungs. This method provides a continuous and relatively accurate measurement of CO [49,62,63] during CPET and can be used to monitor changes in CO over time. Currently, available IGR measurement systems using SF_6_ (i.e., Innocor^®^ CO–Cardiac Output—COSMED) do not include a rhythmic assessment of the exercise ECG, which limits the possibility of monitoring patient safety during CPET-IGR. Of course, this difficulty can be overcome by performing simultaneous ECG recordings with an external system. 

However, there are no guidelines for the inert gas rebreathing method during a cardiopulmonary exercise test (CPET). In addition, there is also a lack of specific standards for assessing the non-invasive measurement of CO during CPET, both for healthy patients and those suffering from diseases and conditions. Hence, there is the need to write a systematic review to bring together the available up-to-date knowledge on using IGR during CPET.

The purpose of this systematic review is to systematically describe the use of IGR for the non-invasive assessment of CO during cardiopulmonary exercise testing, present the extracted data, if applicable, and, based on the information extracted, to identify a proposed CPET report that includes the assessment of CO using the IGR method.

## 2. Methods

### 2.1. Literature Search

This systematic review was performed in accordance with the PRISMA (Preferred Reporting Items for Systematic Reviews and Meta-Analyses) statement [64]. The PICO (Patients, Interventions, Comparisons, and Outcomes) approach was not used, as this systematic review was not to evaluate intervention effects. The following predetermined strategy was used to identify eligible articles. We searched PubMed, Web of Science, Scopus, and Cochrane.

### 2.2. Evidence Acquisition

On 29 December 2022, the targeted online databases were searched. The search query was constructed with the use of MeSH structures: ((“inert gas rebreathing”) AND (“exercise test” OR “walk test” OR “ergometry” OR “stress test” OR “treadmill test” OR “bicycle ergometry test” OR “fitness testing” OR “cardiopulmonary exercise testing” OR “CPET”)). No search filters were applied. The search was completed with a manual search of references from key papers. The primary search returned 261 articles. The number of articles after duplicate removal was narrowed to 186. The initial screening process involved two researchers (AC, ŁM), independently reviewing the titles and abstracts of all the records. Any inconsistencies were discussed until a consensus was reached. Then, the researchers independently screened the full-text articles for inclusion. The data were collected using an Excel spreadsheet. In case of any disagreement, they discussed and reached a consensus on whether to include or exclude the article. The complete protocol is depicted in a PRISMA flowchart (Figure 3).

### 2.3. Inclusion and Exclusion Criteria

Studies were included in the current systematic review if they conformed with the following criteria: (1) a cardiopulmonary exercise test (CPET) had been performed, (2) the inert gas rebreathing method was used in the CO assessment, (3) it included an adult population, and (4) the CPET protocol was described in a precise and explicit manner, allowing the study to be reconstructed. If articles met the predefined criteria, they were included and categorized as eligible. The exclusion criteria included: (1) review articles, (2) studies including a pediatric population, (3) studies including patients with congenital heart diseases, or (4) the entire text was unavailable in English.

### 2.4. Evidence Synthesis and Quality Assessment

Given the experimental nature of the studies and the fact that, in most cases, the IGR method was not the focus of the research, it was not possible to use widely recognized tools for a quality assessment. Therefore, the quality assessment was performed manually by referring to a model study written by Agostoni et al. [57], the only author with defined reference values. The manual evaluation of the quality of the included studies consisted of assessing whether the study description contained an accurately described study protocol, allowing for reproduction. In addition, each study was evaluated against the inclusion criteria (described in Section 2.3) by two independent authors of the systematic review, and any disagreement was resolved by a third author. Reference data presented in the article by Agostoni and co-authors, titled: “Reference values for peak exercise cardiac output in healthy individuals” [57], are presented in Table 1. The study involved 500 normal subjects of varying ages and genders. They underwent a maximal cardiopulmonary exercise test with a peak CO measurement using a specific method. The results showed that peak CO was higher in males compared to females. Both peak CO and peak VO_2_ decreased with age. Moreover, the study provided the formula to predict peak CO from peak VO_2_ values. The equation shows that, in the general population, peak CO = 4.4 × peak VO_2_ + 4.3, while peak CO in males = 4.3 × peak VO_2_ + 4.5 and peak CO in females = 4.9 × peak VO_2_ + 3.6. The data reported in Agostoni’s study are clear, and the study model allows for the reproduction of protocol. Given the lack of established tools for assessing the quality, this approach seemed appropriate for the current systematic review. However, it is important to acknowledge the limitations of this method and the potential for bias in the quality assessment process. Therefore, the results of the quality assessment should be interpreted with caution.

### 2.5. Study Registration

This systematic review has been registered with PROSPERO (International Prospective Register of Systematic Reviews) under registration number CRD42023456820. A detailed protocol for this systematic review is available in PROSPERO, where it can be accessed and evaluated.

### 2.6. Availability of Data

The data, code, and other materials used in this systematic review are available upon request from the corresponding author. We are committed to promoting transparency and reproducibility in research, and we encourage interested parties to reach out for access to the data and materials used in this study.

## 3. Results

### 3.1. Study Selection and Description

The search flowchart is presented in Figure 3.

In Table 2, all included studies were considered, and raw data related to measurements conducted using IGR were extracted wherever feasible (Table 3). The studies included in the systematic review where the extraction of raw data was unattainable are marked with “*” in Table 2. The reasons for the inability to obtain the raw data were data presented solely in graphical form, data presented as value differences, or data presented in units different from those sought in this systematic review. This comprehensive approach ensured a thorough assessment of the available data.

Table 2, presented herein, provides the most critical information, e.g., the protocols used, patients’ characteristics, the study design, and the primary and secondary points. Where feasible, there are comments and conclusions listed. This table serves to streamline and highlight key findings and relevant data points from the studies included, offering a succinct overview of the essential aspects of the research synthesized in this review.

This is the first systematic review to address the topic of a non-invasive CO assessment using IGR during CPET. And as such, we believe it is necessary to provide a comprehensive description of the protocols, indications for cardiopulmonary exercise testing with IGR (CPET-IGR), technical details of the testing, and proposed reporting standards. Due to the lack of a standardized approach to CPET-IGR, the inclusion criteria for this review were broad to capture as many relevant studies as possible.

### 3.2. Participants of the Studies Included

This systematic review resulted in the extraction of 48 articles that matched the search criteria. The total number of patients from all studies is 4465. The average age of the participants ranges from 21 to 72 years old. The populations are diverse in terms of health comorbidities (Section 3.4 and Table 4). For an accurate description of the population of each study, see Table 2.

### 3.3. Protocols Used in CPET-IGR Studies

A cardiopulmonary exercise test typically requires a cycloergometer or a treadmill. The type of exercise equipment used depends on the patient’s preference, physical capabilities, and the test’s purpose. In the studies included in this systematic review, the authors used both cycle ergometers [58,86,88,90] and treadmills [84,87,89,91].

The standard CPET protocol involves the following steps. (1) Pre-test procedures: Before the test, the subject’s height, weight, age, and medical history are recorded. The subject is also instructed on using exercise equipment and wearing an interface for measuring respiratory gases. (2) Baseline measurements: the subject rests quietly for several minutes while measurements of their resting heart rate, blood pressure, and oxygen saturation are taken. (3) Incremental or constant load exercise: The subject begins exercising on a cycle ergometer or treadmill at a low intensity, gradually increasing every minute until exhaustion or a pre-determined endpoint is reached. The exercise protocol can vary based on the individual being tested and the test’s purpose. (4) Respiratory gas measurements: During exercise, the subject wears a mask connected to a metabolic cart that measures the volume of oxygen and carbon dioxide consumed and produced, respectively. These data allow for the various physiological parameters, such as the anaerobic threshold, peak oxygen uptake, and respiratory exchange ratio. (5) ECG monitoring: the subject’s heart rhythm is continuously monitored during exercise using an electrocardiogram (ECG) to detect any abnormalities or changes in the heart rate. (6) Termination of the test: the test is usually stopped when the subject reaches a pre-determined endpoint, such as the maximum heart rate, symptom limitation, or a predetermined workload. (7) Recovery: The subject rests for several minutes while their heart rate, blood pressure, and oxygen saturation measurements are taken. The subject may also be monitored for any post-exercise symptoms or complications. The protocol for CPET-IGR may vary depending on the patient’s specific medical history and condition, and may be adjusted accordingly by the healthcare provider performing the test (Table 1). 

The researchers conducting the studies included in this systematic review conducted CPET-IGR using varied testing protocols. Most of these were tests involving a progressive work rate [31,59,60,61,63,65,66,67,68,69,70,72,73,74,75,76,77,78,79,80,82,83,85,93,94,95,96,97,98,101,102], while a few studies used constant load protocols in addition to progressive work rate tests [71,81,92,99,100]. 

The authors of the papers discussed do not strictly state how the measuring system is connected to the patient (mouthpiece or full-face mask). It should also be noted that there are no studies validating the different interfaces of CPET-IGR.

### 3.4. Health Conditions as Indication for IGR Study

As there are no separate recommendations available for the indications for CPET-IGR, general indications for CPET were used by the authors of the cited papers to perform CPET-IGR. 

According to the American Thoracic Society and American College of Cardiology, there are several common indications for CPET: the diagnosis and assessment of an exercise intolerance in patients with cardiovascular or pulmonary diseases, such as heart failure, chronic obstructive pulmonary disease (COPD), or pulmonary hypertension; the assessment of preoperative risk in patients undergoing major surgery; the determination of exercise-induced bronchoconstriction in patients with suspected exercise-induced asthma; the evaluation of the fitness and training status in athletes or individuals participating in endurance sports; an unexplained dyspnea diagnosis; or the assessment of interstitial lung diseases [103]. 

Among the studies included in this systematic review, the predominant indications for CPET in patients were HF (53% of included articles) and a physical capacity assessment in healthy individuals or athletes (28%). Only 19% of the included articles concerned other patient populations. For instance, only two articles (4% of included articles) focused on COPD patients (Table 4). 

### 3.5. Other

The IGR method has high repeatability, reproducibility, and its reliability is not compromised by atrial fibrillation or pulmonary diseases [45,51,55,56,63,76,84]. The IGR method is, therefore, ideally suitable for determining CO in LVAD (Left Ventricular Assist Device) patients at rest and during exercise. It allows the evaluation of the physical fitness of a particular population of patients. Since the method is non-invasive, it can be performed any number of times, allowing the long-term documentation of hemodynamics [104].

### 3.6. Proposed IGR Report Style

The proposed CPET-IGR report is presented in Table 5. 

## 4. Discussion

The systematic review conducted on the topic of a non-invasive CO measurement using the IGR method during cardiopulmonary exercise testing (CPET) has revealed important insights into the current state of research in this field. While the application of IGR holds promise for assessing cardiac function in a non-invasive manner, the synthesis of findings across numerous studies has highlighted several key challenges that need to be addressed for this method to reach its full potential. 

### 4.1. The Validation and Utility of CPET-IGR

This systematic review encompasses a diverse array of studies exploring the potential of IGR as a valuable tool for assessing CO during CPET. By synthesizing findings from studies across various clinical contexts, the discussion sheds light on the evolving landscape of CO measurement, with an emphasis on the insights provided by the IGR method. 

A particularly important study conducted by Middlemiss et al. [56] reveals the potential of CPET-IGR. Normative data highlight age-related declines in CO and SV, particularly in males. Body size, assessed through BMI (Body Mass Index), significantly impacts CO and SV, especially in younger individuals. IGR proves sensitive to physiological changes, highlighting posture-induced CO shifts. Although limitations exist, this study paves the way for understanding age, body size, and posture effects on CO, offering insights for future investigations and clinical applications.

Several studies acknowledge the variability, validation, and limitations inherent in different CO assessment methods [56,59,70,81,86,91,93,94]. Articles compare the CO measurement methods during CPET, obtained by Innocor (IGR) and other methods including the Fick method, Physioflow (impedance cardiography), Nexfin (pulse contour analysis), finger photoplethysmography, cardiac magnetic resonance and bioreactance. The conclusions from the above studies suggest that the IGR method offers a straightforward, precise, and consistently replicable non-invasive approach for quantifying CO during CPET [56,59,70,91]. However, as stated by Siebenmann et al. [94] and Okwose et al. [70], the four extensively employed and valid techniques for assessing CO during exercise yield notably divergent measurements. Consequently, the choice of method significantly impacts the determination of CO during exercise, and those methods cannot be used interchangeably. 

The systematic review includes, among others, studies focused on patients with heart failure (CHF), a population characterized by complex hemodynamic responses during exercise. The studies demonstrate the potential of IGR to provide valuable insights into the management of HF. A study conducted by Halbirk et al. [31] examines the effects of a 48 h infusion of glucagon-like peptide-1 in patients with compensated chronic heart failure and highlights the potential of IGR to elucidate cardiovascular and metabolic effects in this context. What is more, there are several articles investigating the management of patients with left ventricular assist devices (LVADs) [69,71,76,77,83,87]. Researchers highlight the usefulness of IGR in assessing CO changes and optimizing device settings. This application underscores the role of IGR as a non-invasive tool in the evolving landscape of LVAD care and optimization. Another notable theme in the discussion is the exploration of the predictive power of CO measurements, particularly in relation to exercise capacity and prognosis. Among the studies included in this systematic review, only a few articles addressing the predictive value of the IGR were identified [58,82,83,88,90]. In the Pastormerlo et al. [90] study, IGR was among other methods used in stratifying the risk of HF progression. Goda et al. [58] stated that CO at 25 W, measured non-invasively during submaximal exercise, may have potential value as a predictor of outcomes in patients with CHF. Additionally, Lang et al. [82] describe peak cardiac power, measured non-invasively, as an independent predictor of an outcome that can enhance the prognostic power of peak VO_2_ in the evaluation of patients with HF. However, Y. Shen et al. [88] concluded that peak CPO is not a predictor of cardiac death in Chinese CHF patients. 

According to Jakovljevic et al. [83], powerful exercise-derived prognostic indicators, including peak VO_2_, AT, circulatory power, and the ventilatory efficiency slope, demonstrate limited capacity to reflect the cardiac organ function in patients treated with LVADs.

Hence, further research should focus on investigating the predictive value of the IGR method during CPET. The above-described studies on the prognostic and predictive value mainly focus on HF patients. Data are needed to provide the reliability, reproducibility, and predictive value of the IGR method in patients with conditions other than HF. Special attention should be given to patients suffering from lung diseases.

### 4.2. Lack of Uniform Result Presentation

One of the foremost observations arising from the systematic review is the lack of standardized results presentations across studies utilizing the IGR method. The absence of a consistent reporting format hampers the comparability of results, making it challenging to draw meaningful conclusions from the accumulated data. This inconsistency impedes one’s ability to discern trends, potential variations, or establish normative values for CO measurements. 

### 4.3. Diverse Study Protocols

Another noteworthy finding is the substantial heterogeneity in the study protocols employed in investigations utilizing IGR for CO measurement. The diversity in experimental designs, including variations in the use of masks or mouthpieces, as well as interruptions in CPET for the transition between measurement devices, introduces confounding factors that could influence study outcomes. In particular, the practice of switching from a CPET mask to an IGR mouthpiece during peak exercise phases may introduce variability in the peak values, thereby impacting the reliability of the results obtained. 

### 4.4. Inadequate Methodology Reporting

The systematic review also highlights the inconsistency in reporting the methodology of IGR measurements. Often, authors fail to detail whether a mask or a mouthpiece was used for IGR, which undermines the transparency and reproducibility of the studies. This lack of transparency poses challenges in interpreting the results and prevents the establishment of standardized best practices for IGR utilization. 

### 4.5. Points for Clinical Practice

Considering these challenges, it is evident that there is an urgent need for the development of guidelines and standardized protocols for non-invasive CO measurement using the IGR method during CPET. By establishing clear guidelines on the instrumentation, experimental setup, and result reporting, researchers can enhance the consistency and reliability of their findings. This will enable more meaningful comparisons across studies, facilitating the accumulation of robust evidence for clinical applications. 

### 4.6. Establishing Normative Ranges

Moreover, the systematic review underscores the necessity of generating comprehensive normative ranges for CO measurements using the IGR method. These normative values will serve as a valuable reference point, aiding in the interpretation of results obtained from different populations and under various conditions. Developing a wide range of normative values will contribute to the broader clinical utility of CPET-IGR. In conclusion, while the non-invasive measurement of CO using the IGR method during CPET holds significant potential, the systematic review illuminates the current challenges and limitations. Addressing the issues of result presentation, diverse study protocols, inadequate reporting, and the need for guidelines and normative ranges is imperative for the method’s advancement. By collaboratively addressing these challenges, researchers can unlock the true clinical and research potential of non-invasive CO assessment using IGR during CPET.

## 5. Limitations

The presented systematic review includes several limitations. Firstly, some of the studies included in this review were based on small patient groups [31,51,59,66,76,79], and most of the studies focused on patients with either healthy conditions or heart failure. There were few studies available that focused on other disease entities [85,89,95,96,97,98,99]. Additionally, not all the required values could be obtained from the studies because they were not described. In addition, in most studies, a non-invasive CO assessment using the IGR method was not the main subject under investigation and was only one of many measures used in the study. These limitations should be considered when interpreting the results of this systematic review. Further studies with larger and more diverse patient populations are needed to provide more comprehensive information on the topic.

## 6. Conclusions

Given that the investigators conducting the studies included in this systematic review identified heart failure or a physical capacity assessment in healthy patients as the main indications for CPET and a non-invasive CO assessment, it is reasonable to conclude that the IGR technique is not used in a wide range of patients. Therefore, there is a need to validate and expand research on the IGR technique in patients with various conditions.

In conclusion, this systematic review provides a comprehensive overview of the use of IGR for a non-invasive assessment of CO during CPET. The evidence supports the reliability and validity of IGR as a method for measuring CO during exercise. However, there is a need for standardized protocols, guidelines, and reporting standards to ensure the consistent and accurate use of IGR in CPET assessments. Future research should focus on establishing normative values, and validating quality assessment tools specific to IGR studies. By addressing these gaps, we can enhance the clinical utility of IGR and its contribution to our understanding of cardiovascular function during exercise.

## Figures and Tables

**Figure 1 jcm-12-07154-f001:**
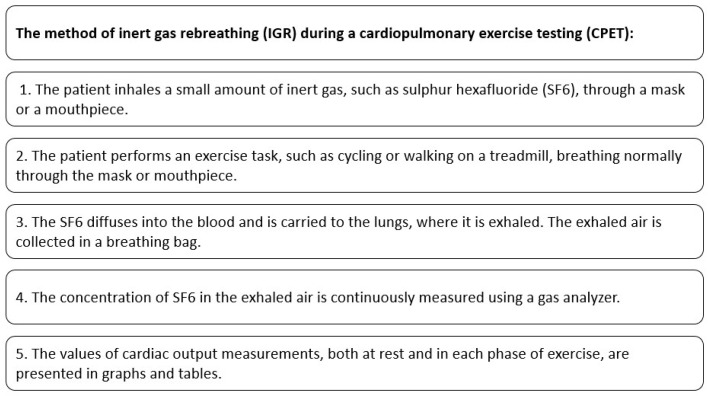
Cardiac output measurement with the use of IGR method during CPET.

**Figure 2 jcm-12-07154-f002:**
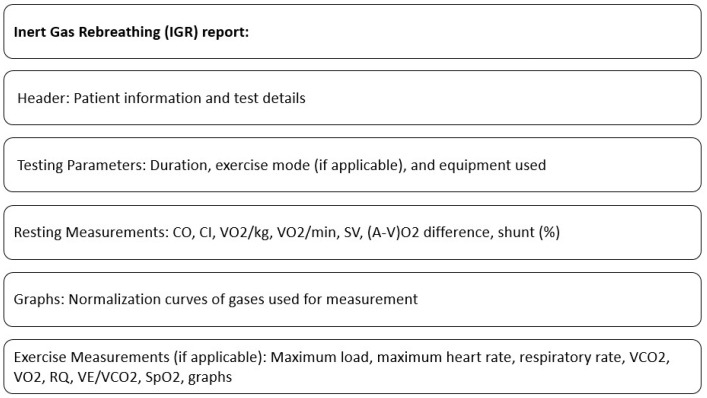
Contents of standard IGR report.

**Figure 3 jcm-12-07154-f003:**
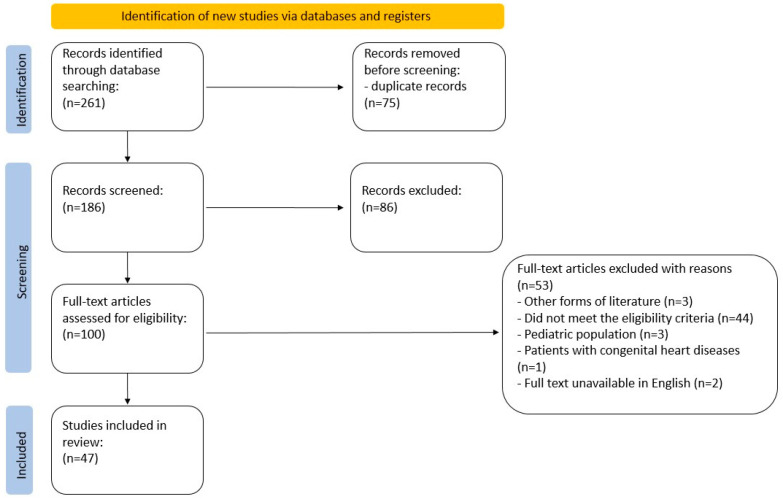
PRISMA flowchart.

**Table 1 jcm-12-07154-t001:** Reference values for peak exercise cardiac output in healthy individuals by Agostoni et al. [57].

		Peak VO_2_ (mL/min)	Peak CO (L/min)	Peak Δ(a-v) (mL/100 mL)	Peak HR (bpm)	Peak SV (mL)	Peak CI (L/min/m^2^)
Total population	All (*n* = 500)	2025 ± 668	13.2 ± 3.5	15.2 ± 2.7	157 ± 19	84.5 ± 21.6	7.33 ± 1.59
	M (*n* = 260)	2494 ± 560	15.3 ± 3.4	16.5 ± 2.7	158 ± 20	96.7 ± 20.3	7.87 ± 1.69
	F (*n* = 240)	1518 ± 309	11 ± 2.1	13.8 ± 2	156 ± 18	71 ± 13.7	6.75 ± 1.24
Age ≤ 40	All (*n* = 181)	2175 ± 688	14.4 ± 3.4	15 ± 2.5	168 ± 14	86.2 ± 19.9	8.15 ± 1.46
	M (*n* = 88)	2735 ± 532	16.9 ± 2.9	16.3 ± 2.5	170 ± 16	100.1 ± 16.9	8.82 ± 1.57
	F (*n* = 93)	1646 ± 277	12.1 ± 1.8	13.7 ± 1.8	167 ± 13	73 ± 12.2	7.52 ± 1.02
Age 41–60	All (*n* = 242)	2042 ± 655	13.1 ± 3.4	15.5 ± 2.8	155 ± 17	85.1 ± 22.6	7.13 ± 1.41
	M (*n* = 134)	2485 ± 515	15.1 ± 3.1	16.7 ± 2.8	156 ± 18	96.7 ± 21.2	7.64 ± 1.45
	F (*n* = 108)	1492 ± 292	10.7 ± 1.9	14 ± 2	153 ± 17	70.5 ± 14.4	6.49 ± 1.05
Age > 60	All (*n* = 77)	1627 ± 483	10.8 ± 2.7	15.1 ± 3	139 ± 20	78.9 ± 21.5	6.04 ± 1.37
	M (*n* = 38)	1969 ± 392	12.2 ± 2.9	16.5 ± 2.9	139 ± 22	89.1 ± 22.5	6.49 ± 1.51
	F (*n* = 39)	1286 ± 283	9.5 ± 1.7	13.7 ± 2.4	140 ± 17	68.8 ± 14.9	5.59 ± 1.06

**Table 2 jcm-12-07154-t002:** Results of the systematic review.

Author and Year	Center	N of Patients	Methods Used	Protocol	Disease/State of Health	Age	Male Gender (n (%))	BMI (kg/m^2^)	Study Design	Primary Endpoint	Secondary Endpoint	Comments and Conclusions
Koshy A. et al., 2019 [65]	Newcastle University and Newcastle upon Tyne Hospitals, Newcastle upon Tyne, UK	33	IGR	**Test:** progressive work rate; **Equipment:** semi-recumbent cycle ergometer; **Watt range/protocol:** 0 to maximum with workload increased at the rate of 10 watts per minute; **Cadence:** 60–70 rpm; **IGR measurement:** at rest and approximately 30 s before maximal exercise; **Other:** respiratory exchange ratio ≥1.05 was used as an objective indicator that a patient is approaching near maximal exertion and when rebreathing maneuver was initiated.	HF	64 ± 9	26 (79%)	28.0 ± 5.4	prospective observational	Association between heart rate variability and hemodynamic response to exercise in chronic heart failure	IGR as secondary investigation	Cardiac autonomic function is not good indicator of overall function and pumping capability of the heart in chronic heart failure.
Fontana P et al., 2011 [66]	University of Zurich, Zurich, Switzerland	15	IGR	**Test:** graded cycling exercise test (GXT) + Wingate test (WT); **Equipment:** cycle ergometer; **Watt range/protocol:** 100 W (males) or 70 W (females) to maximum; **Cadence:** freely chosen (≥70 min^−1^); **IGR measurement:** 130 W and right before volitional exhaustion (“peak exercise”); **Other:** -	none (healthy and non-smoking)	26.7 ± 5.6	15 (100%)		prospective observational	Effect of Wingate test (WT) on cardiac function in comparison with graded cycling exercise (GXT)	IGR as secondary investigation	Single WT produces a hemodynamic response which is characterized by similar cardiac output, higher stroke volume. and lower heart rate compared to peak exercise during a GXT.
Corrieri N et al., 2021 [67]	Centro Cardiologico Monzino, IRCCS, Milan, Italy	496	IGR	**Test:** progressive work rate; **Equipment:** cycle ergometer; **Watt range/protocol:** personalized; **Cadence:** -; **IGR measurement:** at rest and after 4–5 min of loaded cycling (mid-exercise) and at peak exercise; **Other:** -	231 HF patients (HF), 265 healthy volunteers (HV)	45 33.5–55 (HV) 67 57–73 (HF)	144 54% (HV) 169 73% (HF)	23.5 21.4–25.4 (HV) 25.7 23.4–28.4 (HF)	retrospective analysis	Cardiac output changes during exercise in heart failure patients: focus on mid-exercise	IGR as secondary investigation	Mid-exercise VO_2_ and CO portend peak exercise values and identify severe HF patients. Their evaluation could be clinically useful.
Shelton R et al., 2010 [68]	Castle Hill Hospital, University of Hull, Kingston-upon-Hull, UK	72	IGR	**Test:** progressive work rate; **Equipment:** cycle ergometer; **Watt range/protocol:** 15–60 W; **Cadence:** -; **IGR measurement:** at rest and at 15, 30, 45, 60 W; **Other:** -	23 HF patients (HF), 42 patients without HF (no-HF). Seven subjects (four CHF patients and three non-CHF patients with similar demographics) were unable to perform an adequate rebreathing maneuver and were excluded from further analysis	68.2 ± 8.1	95.7%—HF; 61.9%—non-HF	27.6 (7.7)—HF; 27.1 (4.7)—non-HF	prospective observational	Mechanisms of exercise limitation in patients with chronic heart failure	IGR as secondary investigation	During submaximal exercise, patients with systolic heart failure are able to increase their CO to a similar extent as those without with equal levels of oxygen consumption, but requiring a much greater degree of tissue oxygen extraction.
del Torto A et al., 2019 [59]	University of Brescia, Brescia, Italy	7	IGR (CO IN) and modified cardio-impedance (CO PF)	**Test:** progressive work rate; **Equipment:** cycle ergometer; **Watt range/protocol:** 50–250 W; **Cadence:** 80–85 rpm; **IGR measurement:** after 5 min of exercise at each step; **Other:** approximately one minute before exhaustion, the mouthpiece of the metabolic cart was removed and $\dot{Q}$ IN was assessed during the last 30 s of exercise before exhaustion.	none (young, active and healthy, non-smokers)	25 ± 1	7 (100%)		prospective observational	CO measured by bioimpedance by PF vs. IGR by Innocor	__	CO PF seems to represent a valuable alternative to invasive methods for assessing CO during sub-maximal exercise. The CO PF underestimation with respect to CO IN during supra-maximal exercise suggests that CO PF might be less optimal for supra-maximal intensities.
Halbirk M et al., 2010 [31]	Aarhus University Hospital, University of Copenhagen, Copenhagen, Denmark	15	IGR	**Test:** progressive work rate; **Equipment:** cycle ergometer; **Watt range/protocol:** stages lasting 1 min and in increments of 10 watts/min; **Cadence:** -; **IGR measurement:** at rest and at peak exercise; **Other:** -	HF due to ischemic heart disease; GLP-1 infusion group, Placebo group	61 ± 3	13 (87%)	26 ± 3	a double-blinded placebo-controlled crossover design	Cardiovascular and metabolic effects of 48 h glucagon-like peptide-1 infusion in compensated chronic patients with heart failure	IGR as secondary investigation	Short-term GLP-1 treatment has no significant cardiovascular effects in patients without diabetes with compensated HF.
Schmidt T et al., 2018 [69]	German Sports University Cologne, Cologne, Germany	20	IGR	**Test:** progressive work rate; **Equipment:** cycle ergometer; **Watt range/protocol:** Flat ramp exercise protocol was selected. Following a 2 min warm-up phase (10 watts), the load was continually increased (10 watts increase per minute) until patients reached a symptom-induced maximum exercise capacity; **Cadence:** -; **IGR measurement:** Determined individually for each patient based on his CPET results. For example, if in the CPET a patient could achieve a maximum of 75 watts, the steps for inert gas rebreathing were set at 20, 40, and 60 watts. If, however, a patient was already at his maximum performance in the ramp protocol after 60 watts, the steps were correspondingly set to 15, 30, and 45 watts; **Other:** -	HF supported by LVAD	60.8 ± 7.3	10 (100%)	25.7 ± 3.3	prospective observational	Changes in Total Cardiac Output and Oxygen Extraction During Exercise in Patients Supported with an HVAD Left Ventricular Assist Device	IGR as secondary investigation	A significant increase in total CO between rest and sub-maximum exercise level could be observed
Okwose N et al., 2017 [70]	Newcastle University, Newcastle upon Tyne	20	IGR, bioreactance (BR)	**Test:** progressive work rate; **Equipment:** cycle ergometer; **Watt range/protocol:** six steady-state stages each lasting 3 min (30, 60, 90, 120, 150, and 180 W); **Cadence:** -; **IGR measurement:** at rest and at peak exercise; **Other:** rebreathing was performed and lasted about 8–12 s with a gas volume of 40–60% of the participants’ predicted vital capacity.	none (healthy)	32 ± 10	9 (45%)		observational study	CO measured by bioreactance vs. IGR	__	Bioreactance and inert gas rebreathing methods show acceptable levels of agreement for estimating cardiac output at higher levels of metabolic demand. However, they cannot be used interchangeably due to strong disparity in results at rest and low-to-moderate exercise intensity.
Apostolo A et al., 2018 [71]	Centro Cardiologico Monzino, IRCCS, Milan, Italy	33	IGR	**Test:** progressive work rate, constant workload exercise; **Equipment:** cycle ergometer; **Watt range/protocol:** step 1, maximal CPET performed at LVAD pump speed 3; step 2, 2 maximal CPETs, randomly performed, with LVAD pump speed set at 3 or changing from 3 to 5; step 3, 2 constant workload CPETs on 2 different days with LVAD pump speed randomly set at 2 or 4; **Cadence:** -; **IGR measurement:** during constant workload exercise, at rest, and at the end of the active workload exercise; **Other:** the constant workload CPETs were performed at 60% of the workload, reached at the maximal baseline CPE, and lasted for 6 min.	HF patients supported with Jarvik 2000 LVADs	62.4 ± 8.2	32 (97%)		prospective cohort study	Comprehensive effects of left ventricular assist device speed changes on alveolar gas exchange, sleep ventilatory pattern, and exercise performance	IGR as secondary investigation	Short-term LVAD speed increase improves exercise performance, CO, O_2_ kinetics, and muscle oxygenation.
Del Torto A et al., 2017 [72]	Centro Cardiologico Monzino, IRCCS, Milan, Italy	278	IGR	**Test:** progressive work rate; **Equipment:** cycle ergometer; **Watt range/protocol:** personalized ramp protocol aimed at achieving peak exercise in around 10 min; **Cadence:** -; **IGR measurement:** at rest and at peak exercise, determined when they approached maximal exercise, allowing the final 30 s for the rebreathing maneuver; **Other:** -	HF	69 57–74	215 (77%)	25.8 23.4–28.4	retrospective analysis	Contribution of central and peripheral factors at peak exercise in heart failure patients with progressive severity of exercise limitation	IGR as secondary investigation	Peak VO_2_ is strictly and directly related to peak CO in HF patients. In patients with more compromised exercise capacity, there was a lower peak exercise Δ(a-v)O_2_. Moreover, estimation of peak CO from peak VO_2_ in patients with most severe HF is questionable.
Vignati C et al., 2019 [73]	Centro Cardiologico Monzino, IRCCS, Milan, Italy	181	IGR	**Test:** progressive work rate; all patients underwent two CPETs with (CPET + CO) and without CO determination; **Equipment:** cycle ergometer; **Watt range/protocol:** personalized ramp protocol; **Cadence:** -; **IGR measurement:** at rest, at submaximal exercise (usually = 40% of exercise), and at peak; **Other:** -	HF	64.6 ± 11.2	154 (85%)	26.2 ± 4.1	retrospective analysis	Impact of IGR on the main cardiopulmonary parameters during CPET	__	The similar anaerobic threshold and peak oxygen uptake in the two tests with a lower peak workload and higher VE/VCO_2_ slope at CPET CO suggest higher respiratory work and consequent demand for respiratory muscle blood flow secondary to the ventilatory maneuvers. Accordingly, VE/VCO_2_ slope and peak workload must be evaluated with caution during CPET CO.
Lang C et al., 2007 [60]	Columbia University, New York	88	IGR	**Test:** progressive work rate; **Equipment:** cycle ergometer; **Watt range/protocol:** exercise began at a workload of 0 W and increased every 3 min by 25 W until symptom-limited maximal exercise was reached; **Cadence:** -; **IGR measurement:** CO measurements were made at the end of the resting period, at 50 W, and at peak exercise; **Other:** patients were instructed to signal approximately 1 min before peak exercise.	HF	54 ± 13	68 (77%)		prospective observational	Ease of IGR during CPET among HF patients	__	Combined metabolic stress testing with inert gas rebreathing can be easily performed in patients with HF
Morosin M et al., 2018 [74]	Centro Cardiologico Monzino, IRCCS, Milan, Italy	170	IGR	**Test:** progressive work rate; **Equipment:** cycle ergometer; **Watt range/protocol:** personalized ramp protocol aimed at achieving peak exercise in around 10 min; **Cadence:** -; **IGR measurement:** at rest and at peak exercise; **Other:** patients were grouped according to the reason (muscular fatigue or dyspnea) that led them to terminate the procedure.	HF (MF—muscular fatigue *n* = 132; D—dyspnea *n* = 38)	MF 65 ± 11.1; D 65.1 ± 11.4	MF 87%; D 82%	MF 26.4 ± 4.3; D 26.1 ± 3.9	retrospective analysis	To determine if patients experiencing dyspnea or muscle fatigue exhibited distinct ventilatory or hemodynamic responses during exercise	IGR as secondary investigation	Exercise performance, hemodynamics, and respiratory pattern do not identify heart failure patients who end exercise with dyspnea from those with fatigue.
Cattadori G et al., 2011 [75]	Centro Cardiologico Monzino, IRCCS, Milan, Italy	70	IGR	**Test:** progressive work rate; **Equipment:** cycle ergometer; **Watt range/protocol:** ramp pattern; **Cadence:** -; **IGR measurement:** at rest and at peak exercise; **Other:** -	HF (“Pre”—before—and “Post”—after—an 8-week training program)	61.6 ± 9.6	54 (77%)		prospective cohort study	Hemodynamic Effects of Exercise Training in Heart Failure	IGR as secondary investigation	Exercise training improves peak VO_2_ by increasing CO with unchanged a-vO_2_ diff
Reiss N et al., 2018 [76]	Bad Rothenfelde, Germany	7	IGR	**Test:** progressive work rate; **Equipment:** cycle ergometer; **Watt range/protocol:** ramp protocol of 10 W/min; **Cadence:** -; **IGR measurement:** based on the results, the test protocol for inert gas rebreathing is then individually deduced; **Other:** In order to compensate for the transfer from ramp to step protocol, we subtract 15–20% of the maximum watt level achieved during CPET. The result is then split into three equal steps.	HF supported by LVAD	51.7 ± 7.6	7 (100%)	26.3 ± 2.5	prospective observational	IGR as a helpful tool in the management of left ventricular assist device patients	__	The inert gas rebreathing method is ideal for determining cardiac output in LVAD patients at rest and during exercise. It is helpful for estimating physical capacity and providing exercise guidance in this special patient group; nevertheless, until now, it is unjustifiably underused. Inert gas rebreathing displays high reproducibility, and its measuring accuracy is not impacted by either atrial fibrillation or pulmonary diseases.
Vignati C et al., 2017 [77]	Centro Cardiologico Monzino, IRCCS, Milan, Italy	15	IGR	**Test:** progressive work rate; **Equipment:** cycle ergometer; **Watt range/protocol:** -; **Cadence:** -; **IGR measurement:** at rest and at peak exercise; **Other:** on 2 days at LVAD pump speed set randomly at 2 and 4.	HF supported by LVAD	60.8 ± 7.6	15 (100%)		randomized control trial	The impact of adjusting LVAD pump speed in HF patients on CO	IGR as secondary investigation	In HF, an increase in CO with a higher LVAD pump speed is associated with increased peak VO_2_, postponed anaerobic threshold, and improved ventilatory efficiency.
Iellamo F et al., 2013 [78]	a Istituto di Ricovero e Cura a Carattere Scientifico San Raffaele Pisana, Rome, Italy	16	IGR	**Test:** progressive incremental test; **Equipment:** treadmill; **Watt range/protocol:** modified Bruce protocol; **Cadence:** -; **IGR measurement:** -; **Other:** -	postinfarction CHF—chronic heart failure (ACT—aerobic continuous training *n* = 8; AIT—aerobic interval training *n* = 8)	ACT 62.6 ± 9 AIT 62.2 ± 8	16(100%)	ACT 27.2 ± 3 AIT 27.8 ± 2	randomized control trial	To test the hypothesis that aerobic continuous training (ACT) and aerobic interval training (AIT) induce similar effects on functional capacity, central hemodynamics, and metabolic profile in patients with postinfarction CHF	IGR as secondary investigation	ACT and AIT both induce significant improvement in aerobic capacity in patients with postinfarction CHF, without significant differences between the two training modes.
Pokan R et al., 2014 [79]	University of Vienna, Vienna, AUSTRIA	8	IGR	**Test:** incremental pretest and continuous 24 h ultra-endurance performance; **Equipment:** cycle ergometer; **Watt range/protocol:** blood lactate concentration was used to assess the first and second lactate turn point (LTP1 and LTP2) and the corresponding power output for each subject to subsequently determine the appropriate submaximal workload that the athlete could sustain and complete during the ultra-endurance test; **Cadence:** ≥70 rpm; **IGR measurement:** at rest and during the last 5 min of every hour throughout the 24 h ultra endurance performance; **Other:** one week before the continuous 24 h ultra endurance performance, subjects completed a single incremental cycle ergometer exercise test to volitional fatigue.	ultra endurance cycling athletes	39 ± 8	8 (100%)		prospective observational	Myocardial Dimensions and Hemodynamics during 24 h Ultra endurance Ergometry	IGR as secondary investigation	The decrease in HR during 24 h of ultra-endurance exercise was due to hypervolemia and the associated ventricular loading, increasing left ventricular diastolic dimensions.
Accalai E et al., 2021 [61]	Centro Cardiologico Monzino, IRCCS, Milan, Italy	1007	IGR	**Test:** progressive work rate; **Equipment:** cycle ergometer; **Watt range/protocol:** personalized ramp protocol; **Cadence:** -; **IGR measurement:** at rest, at submaximal exercise (50% of exercise duration), and at peak exercise; **Other:** -	HF—heart failure (*n* = 507), HC—healthy controls (*n* = 500)	HC 44.9 ± 13.4; HF 65.3 ± 12.4	HC 240 (48%); HF 400 (81%)	HC 23.4 ± 3.4; HF 26 ± 4.2	retrospective analysis	Non-invasive estimation of stroke volume during exercise from oxygen in heart failure patients	IGR as secondary investigation	In heart failure patients, both the estimation of stroke volume and the measurement of oxygen pulse during exercise are indicative of stroke volume. However, there is notable variability in individual data, making them suitable for population studies, but not sufficiently reliable for assessing a single subject. As a result, direct measurement of stroke volume is necessary when accuracy is required.
Fontana p et al., 2010 [80]	University of Zurich, Zurich, Switzerland	16	IGR	**Test:** progressive work rate (3 tests); **Equipment:** cycle ergometer; **Watt range/protocol:** test 1—increments of 30 W every 2 min until volitional exhaustion; **Cadence:** freely chosen (≥70 rpm); **IGR measurement:** in test 2 and 3 at 69, 77, and 85% peak power attained in test 1; **Other:** Test 1 served for determining peak power and maximal gas exchange. The respective rebreathings were performed in a randomized order, either during the first or the second minute on each stage (either in test 2 or 3).	recreationally trained male cyclists	25.4 ± 2.9	16 (100%)		prospective observational	Non-invasive hemodynamic assessments using InnocorTM during standard graded exercise tests (GXT)	__	It is feasible to non-invasively determine CO by IGR at 46 and 103 s on moderate and high-intensity submaximal GXT stages, using the InnocorTM device. This study also shows that CO of recreationally trained cyclists does not change between these two points in time at 69, 77, and 85% peak power.
Prochnau D et al., 2012 [81]	Friedrich Schiller University, Jena, Germany	27	IGR, impedance cardiography	**Test:** constant workload exercise; **Equipment:** cycle ergometer; **Watt range/protocol:** standardized, steady-state bibycle exercise at 30 W; **Cadence:** -; **IGR measurement:** at rest was measured with the following AVD: 60, 90, 120, and 150 ms consecutively; **Other:** -	HF + CRT	69.9 ± 10	22 (81.5%)		prospective observational	Optimization of the atrioventricular delay during cardiac resynchronization therapy (CRT) using a device for non-invasive measurement of cardiac index (CI) at rest and during exercise	IGR as secondary investigation	Shortening or lengthening of the atrioventricular delay (AVD). AVD during exercise has no impact on CI in CRT patients. On the basis of our results, we conclude that in CRT patients the AVD should be programmed and fixed even during exercise.
Lang C et al., 2009 [82]	Columbia University, New York, NY	148	IGR	**Test:** progressive work rate; **Equipment:** cycle ergometer; **Watt range/protocol:** exercise began at a workload of 0 W and increased every 3 minutes by 25 W until symptom limited maximum; **Cadence:** -; **IGR measurement:** at the end of the rest period, at 25 W, and at peak exercise; **Other:** patients were instructed to signal 1 min before peak exercise.	HF	53 ± 14	119 (80%)	28 ± 5	prospective observational	To compare the prognostic value of peak CO and cardiac power to peak VO_2_ in chronic heart failure patients	IGR as secondary investigation	Peak cardiac power, measured non-invasively, is an independent predictor of outcome that can enhance the prognostic power of peak VO_2_ in the evaluation of patients with heart failure.
Agostoni P et al., 2017 [57]	Centro Cardiologico Monzino, IRCCS, Milan, Italy	500	IGR	**Test:** progressive work rate; **Equipment:** cycle ergometer; **Watt range/protocol:** personalized ramp protocol aimed at achieving peak exercise in 8 to 12 min; **Cadence:** -; **IGR measurement:** at rest and at peak exercise, determined when they approached maximal exercise, allowing the final 30 s for the rebreathing maneuver; **Other:** -	none (healthy)	45 ± 13.5	260 (52%)		prospective observational	To establish the reference Values for Peak Exercise Cardiac Output in Healthy Individuals	__	The simultaneous measurement of CO and VO_2_ at peak exercise in a large sample of healthy subjects provided an equation to predict peak CO from peak VO_2_ values.
Jakovljevic D et al., 2011 [83]	Newcastle University, Newcastle upon Tyne	54	IGR	**Test:** progressive work rate; **Equipment:** treadmill; **Watt range/protocol:** modified Bruce protocol (implanted LVAD) or the Bruce protocol (explanted LVAD); **Cadence:** -; **IGR measurement:** At rest and at peak exercise. Patients were instructed to give a 1 min warning before they felt they would end the exercise so that a final cardiac output rebreathing measurement was obtained; **Other:** -	HF—heart failure (*n* = 20); LVAD—implanted LVAD (*n* = 18); exLVAD—explanted LVAD (*n* = 16)	22–64	54 (100%)	26.8 ± 4.7	prospective observational	To assess the relationship between cardiac pumping capability represented by peak cardiac power output and peak oxygen consumption, anaerobic threshold, ventilatory efficiency slope, and peak circulatory power in patients treated with LVAD	IGR as secondary investigation	Powerful exercise-derived prognostic indicators, including peak oxygen consumption, anaerobic threshold, circulatory power, and ventilatory efficiency slope demonstrate limited capacity to reflect cardiac organ function in patients treated with LVADs. The imperfect correlation between cardiac power and other exercise-derived variables may suggest that exercise performance of LVAD-implanted patients may be limited by factors other than cardiac.
Fontana P et al., 2009 [63]	University of Zurich, Zurich, Switzerland	30	IGR	**Test:** progressive work rate (2 sessions, 4 tests—both, sessions A and B, consisted of 2 graded exercise tests (A1 and A2; B1 and B2) with 1 h rest in between); **Equipment:** cycle ergometer; **Watt range/protocol:** start at 100 W (males) or 70 W (females) to volitional exhaustion; **Cadence:** freely chosen (≥70 rpm); **IGR measurement:** at rest, at 130 W, and right before volitional exhaustion (“peak exercise”); **Other:** -	healthy, asymptomatic, non-smoking, and recreationally active	30.6 ± 4.5	15 (50%)		prospective observational	Reliability of Measurements with Innocor TM during Exercise	__	Innocor™ (Denmark, Odense) delivers safe and reliable measurements of cardiac output, gas exchange, and ventilation. Therefore, Innocor™ can be used to assess these parameters in exercise physiology studies as well as in performance diagnostics.
Jakovljevic D et al., 2012 [84]	Newcastle University, Newcastle upon Tyne	19	IGR	**Test:** progressive work rate; **Equipment:** treadmill; **Watt range/protocol:** modified Bruce protocol; **Cadence:** -; **IGR measurement:** at rest and at near-maximal exercise [defined as at least one of the following: RER (respiratory exchange ratio) > 1.05, the absence of a rise in VO_2_ with further increase in exercise intensity, Borg scale > 17]; **Other:** -	HF—heart failure	62 ± 11	15 (79%)	26.6 ± 4.0	prospective observational	Reproducibility of cardiac power output and other cardiopulmonary exercise indices in patients with chronic HF	__	Cardiac power output demonstrates good reproducibility suggesting that there is no need for performing more than one CPET. As a direct measure of cardiac function (dysfunction) and an excellent prognostic marker, it is strongly advised in the assessment of patients with chronic HF undergoing CPET.
Okwose N et al., 2019 [51]	Newcastle University, Newcastle upon Tyne	13	IGR	**Test:** progressive work rate; **Equipment:** semi-recumbent cycle ergometer; **Watt range/protocol:** 6 steady-state stages, each lasting 3 min (30, 60, 90, 120, 150, and 180 W); **Cadence:** -; **IGR measurement:** at rest and at 60, 120, 150, and 180 W; **Other:** participants visited the laboratory on 2 occasions (2 days apart, Test 1 and Test 2).	healthy	27 (23–32)	10 (77%)	23.5 ± 2.2	prospective observational	Reproducibility of IGR Method to Estimate CO at Rest and During CPET	__	The IGR method demonstrates an acceptable level of test–retest reproducibility for estimating CO at rest and during CPET at higher metabolic demands.
Vignati C et al., 2021 [85]	Centro Cardiologico Monzino, IRCCS, Milan, Italy.	115	IGR	**Test:** progressive work rate; **Equipment:** cycle ergometer; **Watt range/protocol:** -; **Cadence:** -; **IGR measurement:** at rest and at peak exercise; **Other:** -	patients with PMR (primary mitral regurgitation, *n* = 62) and FMR (functional mitral regurgitation, *n* = 53), before (PRE-pMVR) and after (POST-pMVR) pMVR (percutaneous edge-to-edge mitral valve repair)	77 ± 8	75 (65%)	25 ± 4	prospective observational	To assess the efficacy of percutaneous edge-to-edge mitral valve repair (pMVR) at rest by echocardiography, VO_2_, and CO (IGR) measurement and during CPET with CO measurement	IGR as secondary investigation	The data confirm pMVR-induced clinical improvement and reverse ventricular remodeling at a 6-month analysis and show, despite an increase in CO, an unchanged exercise performance, which is achieved through a ‘more physiological’ blood flow distribution and O_2_ extraction behavior. Direct rest and exercise CO should be measured to assess pMVR efficacy.
Bentley R et al., 2019 [86]	Queen’s University, Kingston, ON, Canada	22	IGR, finger photoplethysmography	**Test:** progressive work rate; **Equipment:** cycle ergometer; **Watt range/protocol:** Test beginning with rest and increasing by 40 W every 4 min until 160 W. Beyond 160 W, exercise increased by 25 W every minute until volitional exhaustion; **Cadence:** self-selected; **IGR measurement:** during the last ~30 s of rest and each completed exercise intensity up to 160 W; **Other:** -	none (healthy) PRE—before training; POST—after training	21 ± 2	22 (100%)	23.9 ± 2.5	prospective observational	Importance of cardiac response phenotype	IGR as secondary investigation	The findings emphasize the importance of recognizing individual response differences in attempting to understand the integrated cardiovascular response to exercise and to exercise training.
Jakovljevic D et al., 2010 [87]	Newcastle University, Newcastle upon Tyne	12	IGR	**Test:** progressive work rate; **Equipment:** treadmill; **Watt range/protocol:** modified Bruce protocol; **Cadence:** -; **IGR measurement:** at rest and at peak exercise; **Other:** Patients visited the transplant exercise laboratory twice during the same day with at least 4 h rest between the two visits. During the first (morning) visit, the HeartMate II LVAD support was optimal with speed ranging from 9000 to 9600 revolutions per minute (rpm). Provided that the international normalization ratio was >2 during the second (afternoon) visit, the HeartMate II LVAD support was reduced and the speed decreased from optimal to 6000 rpm.	HF supported by LVAD	33 ± 13	12 (100%)	23.6 ± 4.2	prospective observational	To assess cardiac and exercise performance in patients implanted with the HeartMate II LVAD under two settings: (i) optimal device support and (ii) reduced device support.	IGR as secondary investigation	HeartMate II LVAD can confer both resting and peak cardiac functional benefits to patients with end-stage HF, thus improving exercise capacity.
Shen Y et al., 2016 [88]	Tongji Hospital of Tongji University, Shanghai, China	129	IGR	**Test:** progressive work rate; **Equipment:** cycle ergometer; **Watt range/protocol:** modified Ramp10 protocol—the exercise began with a 3 min warm-up at 0 W, and thereafter, 10 W increment in load was administered every minute after exercising at 20 W for 2 min; **Cadence:** 60–70 rpm; **IGR measurement:** at the end of the rest period, at 30 W, 60 W, 90 W, and 120 W; **Other:** -	CHF—chronic heart failure	59.1 ± 11.4	113 (87.6)	24.7 ± 3.7	prospective observational	The Prognostic Value of Peak CPO in Chinese Patients with Chronic HF	IGR as secondary investigation	Peak CPO is not a predictor of cardiac death in Chinese CHF patients.
Rajani R et al., 2010 [89]	Guy’s and St Thomas’ Hospitals Foundation Trust, London, UK	38	IGR	**Test:** progressive work rate; **Equipment:** treadmill; **Watt range/protocol:** Bruce protocol modified by two warm-up stages; **Cadence:** -; **IGR measurement:** before the commencement of the exercise test, at each stage during exercise, and then during recovery; **Other:** -	aortic stenosis (AS—asymptomatic *n* = 28; S—symptomatic *n* = 10)	63 (29–83)	32 (84%)	26.7 ± 3.4	prospective observational	Relationship between CI and revealed symptoms in patients with aortic stenosis	IGR as secondary investigation	Revealed symptoms on treadmill exercise in apparently asymptomatic aortic stenosis were associated with lower peak myocardial VO_2_ and lower peak stroke index during exercise. The strongest resting predictor of revealed symptoms and of peak cardiac index was the blood BNP level.
Pastormerlo L et al., 2015 [90]	Fondazione G. Monasterio CNR-Regione Toscana, Pisa, Italy	30	IGR	**Test:** progressive work rate; **Equipment:** cycle ergometer; **Watt range/protocol:** ramp protocol individualized to achieve target respiratory exchange ratio of at least 1.05 and exercise duration of 10 ± 2 min; **Cadence:** -; **IGR measurement:** at rest and at peak exercise; **Other:** -	HF (hsT < 20%—hs-Troponin Increase < 20% *n* = 20; hsT > 20%—hs-Troponin Increase > 20% *n* = 10)	62.5 ± 1.5	27 (90%)	25.4 ± 3.2	prospective observational	Usefulness of Highly Sensitive Troponin Elevation After Effort Stress to Unveil Vulnerable Myocardium in Patients With HF	IGR as secondary investigation	The association of troponin release with norepinephrine, CO, and NT-proBNP changes after effort suggests a pathophysiological link among transient hemodynamic overload, adrenergic activation, and myocardial cell damage, likely identifying a clinical subset at greater risk for HF progression.
Goda A et al., 2009 [58]	Columbia University, New York	145	IGR	**Test:** progressive work rate; **Equipment:** upright, braked cycle ergometer; **Watt range/protocol:** after 3 min of rest, exercise was begun at 0 W and increased every 3 min by 25 W until symptom-limited maximal exercise was achieved; **Cadence:** -; **IGR measurement:** at the end of the rest period, at 25 W, and at peak exercise; **Other:** -	HF	51.1 ± 13.6	186 (72%)	28.7 ± 13.5	prospective observational	Usefulness of Non-Invasive Measurement of CO During Sub-Maximal Exercise to Predict Outcome in Patients With Chronic HF	__	CO at 25 W measured non-invasively during submaximal exercise may have potential value as a predictor of outcomes in patients with CHF.
Hassan M et al., 2017 [91]	Cairo University, Egypt	97	IGR; CC—cadriac catheterization; CMR— cardiac magnetic resonance; echocardiography	**Test:** CPET; **Equipment:** treadmill; **Watt range/protocol:** -; **Cadence:** -; **IGR measurement:** -; **Other:** the resting CO measurements were repeated after 15 min to check for reproducibility of measurements in 30 patients.	HF	42 ± 15.5	71 (73.2%)	28 ± 6.4	prospective observational	Validation of Non-invasive Measurement of CO Using IGR in a Cohort of Patients With HF and Reduced Ejection Fraction	__	The IGR method is a simple, accurate, and reproducible non-invasive method for quantification of CO in patients with advanced heart failure. The prognostic value of this simple measurement needs to be studied prospectively.
Lee W et al.m 2011 [92]	Golden Jubilee National Hospital, Glasgow, UK	24	IGR	**Test:** progressive work rate, followed by constant load; **Equipment:** cycle ergometer; **Watt range/protocol:** progressive work rate—2 min of rest, followed by 3 min of unloaded cycling, and then cycling at a ramp rate of 5–20 Watts/min to achieve peak work rate in 8–12 min; constant work—2 min of rest followed by 3 min of cycling at 40% maximal work rate predetermined in the incremental CPET; **Cadence:** -; **IGR measurement:** during constant load cycling, at the end of exercise; **Other:** the development of right-to-left shunt through a patent foramen ovale during exercise was assessed using a set of published pulmonary gas exchange criteria.	precapillary PH (pulmonary hypertension)	59 ± 15	16 (66%)		prospective observational	Use of non-invasive hemodynamic measurements to detect treatment response in precapillary PH	__	Non-invasive IGR hemodynamic measurements could be used to detect treatment response in patients with precapillary PH and may be more responsive to change than 6MWD in fitter patients.
Bentley R et al., 2018 [93]	Queen’s University, Kingston, Ontario, Canada	31	IGR, finger photoplethysmography	**Test:** progressive work rate; **Equipment:** cycle ergometer; **Watt range/protocol:** Test beginning with rest and increasing by 40 W every 4 minutes until 160 W. Beyond 160 W, exercise increased by 25 W every minute until volitional exhaustion; Cadence: self-selected; **IGR measurement:** at rest and during exercise at 40 W increments once every 4 min up to 160 W; **Other:** pilot work (*n* = 4)-CO variability—2.2% for exercise intensities up to 200 W.	none (healthy and non-smoking); divided post-hoc into: LCR—lower cardiac resoinders (*n* = 10); HCR—higher cardiac responder (*n* = 10)	21 ± 3	31 (100%)	23 ± 2.4	prospective observational	Impact of interindividual differences in CO during submaximal exercise on exercising muscle oxygenation and ratings of perceived exertion	IGR as secondary investigation	Interindividual differences in CO during submaximal exercise have no impact on exercising muscle oxygenation and ratings of perceived exertion.
Siebenmann C et al., 2015 * [94]	University of Zürich, Institute of Physiology	12	the Fick method, IGR, impedance cardiography, pulse contour analysis	**Test:** progressive work rate; **Equipment:** cycle ergometer; **Watt range/protocol:** First trial (normoxic)—For 3 min unloaded, 6 min at 112.5 W, and 10 min at 150 W. Thereafter, the workload was increased by 37.5 W every 1.5 min until exhaustion. Second trial (performed with the inspired O_2_ fraction reduced by N2 dilution) to 12% (∼4000 m)—3 min bouts of cycling at 75 W, 112.5 W, and 150 W, respectively. The load was then increased as in the normoxic trial by 37.5 W every 1.5 min until exhaustion; **Cadence:** 75 rpm; **IGR measurement:** In normoxia, CO was determined at rest, at 112.5 W, 150 W, and at every second step of the incremental trial, i.e., every third minute after the 185.5 W workload. During hypoxia, CO was assessed at the same workloads and additionally at 75 W with the rebreathing maneuver starting immediately after blood sampling, approximately 30 s before the end of the workload; **Other:** -	none (healthy)	25 ± 5	12 (100%)		prospective observational	CO during exercise: comparison of four methods	__	Although all methods have been validated, they may generate significantly different CO values within the same subjects. Different measurement techniques for CO should be taken into account by researchers as well as physicians when comparing the outcome of evaluations.
Ananey, O. et al., 2011 * [95]	Trinity College Dublin, Dublin, Ireland	29	IGR	**Test:** progressive work rate; **Equipment:** cycle ergometer; **Watt range/protocol:** On testing day 1, subjects completed a graded cycle test to fatigue. After 3 min of rest, subjects cycled at an initial workload of 40 W for 3 min before the workload was increased by 20 W every 3 min until a cadence of 60 rpm could not be maintained (i.e., task failure). On testing days 2 and 3, subjects were required to complete six 7 min exercise bouts on each day, so that, during the 2 d, each subject completed four exercise bouts at each of the following three intensities relative to the workloads achieved at VT and peak VO_2_, as defined by the graded test performed on day 1:50% VT, 80% VT, and midpoint between VT and peak workload (50% A), **Cadence:** 60 rpm; **IGR measurement:** during the final exercise bout, only on testing day 3, but at two time points (at 30 and 240 s) during each 7 min exercise period at each intensity; **Other:** -	DMT2—type 2 diabetes mellitus (*n* = 9); HC—overweight but otherwise healthy controls (“heavy controls”, *n* = 9); and LC—lean and healthy controls (“lean controls”, *n* = 11)	49.1 ± 5.7 (DMT2); 42.5 ± 12.6 (HV); 44.1 + 8.2 (LC)	0(0%)	34.4 ± 7.4 (DMT2); 29.0 ± 8.7 (HV); 22.7 ± 1.2 (LC)	prospective observational	To investigate whether CO responses are related to VO_2_ kineties during cycling in type 2 diabetes	IGR as secondary investigation	Type 2 diabetes slows the dynamic response of VO_2_ during light and moderate relative intensity exercise in females, but this occurs in the absence of any slowing of the CO response during the initial period of exercise.
Van Doan Tuyet Le, 2015 * [96]	Roskilde University Hospital, Roskilde, Denmark	131	IGR	**Test:** progressive work rate; **Equipment:** cycle ergometer; **Watt range/protocol:** the load was calculated and set to reach the predicted VO_2_ in approximately 8–10 min, with linear increments in load at 1 min intervals after 3 min of unloaded cycling, **Cadence:** -; **IGR measurement:** at rest and during submaximal exercise beyond the anaerobic threshold, which is the point of maximal stroke volume during exercise; **Other:** the stroke volume index (SVI) was calculated at rest and during exercise (SVIrest and SVI-exercise) from the cardiac output, heart rate, and body surface.	Aortic stenosis (asymptomatic or equivocal symptomatic)	72.1 ± 9.3	83 (63%)	26.8 ± 4.0	prospective observational	Cardiopulmonary Exercise Testing in Aortic Stenosis	IGR as secondary investigation	Equivocal symptomatic patients are characterized by lower pVO_2_ and a low aortic valve area index, but with lower gradients. Both CPET and IGR confirmed that this was due to a lower stroke volume.
O’Connor E et al., 2015 * [97]	Trinity College Dublin, Dublin, Ireland	54	IGR	**Test:** progressive work rate; **Equipment:** cycle ergometer; **Watt range/protocol:** Day 1—an initial power output of 40 W for 3 min using a fixed cadence (60 rpm). Thereafter, the power output was increased by 30 W every 3 min until the required cadence could not be maintained. Day 2—subjects performed six, 6 min bouts of cycling at 80% VT, with each bout separated by 12 min of rest and preceded by a 3 min cycling period at 10 W (“unloaded” cycling); **Cadence:** 60 rpm; **IGR measurement:** at rest, 30 s, and 240 s **Other:** -	DMT2—type 2 diabetes mellitus (*n* = 33; 15 middle-aged; 18 older); HC—healthy controls (*n* = 21; 11 middle-aged; 10 older)	(Middle-aged: controls/T2DM; Older: controls/T2DM): (48 ± 10/52 ± 7; 64 ± 2/64 ± 3)	54 (100%)	(Middle-aged: controls/T2DM; Older: controls/T2DM): (28.8 ± 3.8/29.3 ± 2.8; 28.2 ± 2.9/30.3 ± 3.1)	prospective observational	Differential effects of age and type 2 diabetes on dynamic vs. peak response of pulmonary oxygen uptake during exercise	IGR as secondary investigation	The mechanisms by which type 2 diabetes induces significant reductions in peak exercise performance are linked to a slower dynamic response of VO_2_ and reduced systemic vascular conductance responses in middle-aged men, whereas this is not the case in older men.
Galera R et al., 2021 * [98]	Hospital Universitario La Paz-IdiPAZ, Madrid, Spain	82	IGR	**Test:** progressive work rate; **Equipment:** cycle ergometer; **Watt range/protocol:** 15 W/min; **Cadence:** -; **IGR measurement:** at the end of the resting period, at 15 W, and at 45 W; **Other:** -	COPD (*n* = 57), healthy control (HC) (*n* = 25)	COPD 62 ± 10; HC 59 ± 8	COPD 38 (67%); HC 18 (72%)	COPD 26.8 ± 4.0; HC 26.4 ± 3.4	prospective observational	To compare the exercise response of SV and CO between COPD patients with or without dynamic hyperinflation and control subjects	IGR as secondary investigation	Dynamic hyperinflation decreases the cardiac response to exercise of COPD patients.
Joshua H. Jones et al., 2016 * [99]	Queen’s University, Kingston, Ontario, Canada	45	IGR	**Test:** progressive work rate and constant wok rate test (on a different day, at least 48 h apart); **Equipment:** cycle ergometer; **Watt range/protocol:** progressive work rate—selected individually; constant work—5 min 25 and 50 W exercise bouts separated by 5 min resting periods; **Cadence:** -; **IGR measurement:** at rest and after 3 min of exercise; **Other:** -	COPD (*n* = 19), healthy control (HC) (*n* = 26)	COPD 62 ± 6 (male); 65 ± 11 (female); HC 61 ± 9	COPD 9 (47%); HC 14 (54%)	COPD 27.4 ± 5.2; HC 26.8 ± 4.1	prospective observational	Exercise capacity of COPD patients with emphysema	IGR as secondary investigation	The observed association between emphysema extent and low exercise PBF likely represents the combined effects of poor lung perfusion in emphysematous areas with the undesired consequences of higher alveolar dead space on IGR measurements.
Kiely C et al., 2015 * [100]	Trinity College Dublin, Dublin, Ireland	44	IGR	**Test:** progressive work rate, constant work test; **Equipment:** cycle ergometer; **Watt range/protocol:** day 1—subjects completed a graded cycling exercise test; after a 3 min period of seated rest, all subjects began the graded test by cycling at an initial power output of 40 W for 3 min using a fixed cadence (60 rpm). Thereafter, the power output was increased by 20 W every 3 min until the required cadence could not be maintained (i.e., task failure). On testing day 2, subjects performed six, 6 min bouts of cycling at 80% VT, with each bout separated by 12 min of rest and preceded by a 3 min cycling period at 10 W; **Cadence:** 60 rpm; **IGR measurement:** at rest, 30 s, and 240 s; **Other:** -	DMT2—type 2 diabetes mellitus (*n* = 22; 8 premenopausal (DMT2/Pre-m), 11 postmenopausal (DMT2/Post-m)); HC—healthy controls (n= 22; 11 premenopausal (HC/Pre-m), 11 postmenopausal (HC/Post-m))	(DMT2/Pre-m) 44 ± 1; (DMT2/Post-m) 55 ± 4; (HC/Pre-m) 40 ± 5; (HC/Post-m) 55 ± 2	0(0%)	(DMT2/Pre-m) 33.1 ± 5; (DMT2/Post-m) 31.1 ± 3; (HC/Pre-m) 30.4 ± 3.3; (HC/Post-m) 30.3 ± 3.2	prospective observational	Hemodynamic responses during graded and constant-load plantar flexion exercise in middle-aged men and women with type 2 diabetes	IGR as secondary investigation	The magnitude of T2D-induced impairments in peak $\dot{V}$O_2_ and $\dot{V}$O_2_ kinetics is not affected by menopausal status in participants younger than 60 yr of age.
Laroche D et al., 2013 * [101]	University Hospital of Dijon, Dijon, France	18	IGR	**Test:** progressive work rate; **Equipment:** cycle ergometer; **Watt range/protocol:** initial power 50 W/min, increments 25 W/min; **Cadence:** 60 rpm; **IGR measurement:** at rest, at 11 minutes, and at 20 min after the start of exercise; **Other:** -	none (healthy)	27.4 ± 5.3	15 (83%)	22.7 ± 1.8	prospective observational	To assess the safety and acute effects of a procedure using perceived exertion during a prior submaximal concentric (CON) test to individualize eccentric (ECC) cycling exercise intensity	IGR as secondary investigation	Moderate-intensity ECC cycling exercise tailored according to perceived exertion during a prior CON test is well tolerated. It corresponds to a limited muscular use of oxygen and to an isolated increase in stroke volume. It appears to be a feasible procedure for preconditioning before ECC training.
Middlemiss, J. E. et al., 2018 * [56]	University of Cambridge, Cambridge, UK	20	IGR	**Test:** progressive work rate; **Equipment:** cycle ergometer; **Watt range/protocol:** exercise at 20 and 35 rpm, corresponding to approximately 12 and 25 watts, respectively, for 5 min at each workload; **Cadence:** 20–35 rpm; **IGR measurement:** at rest, at 11 min, and in the final minute of each workload; **Other:** -	none (healthy)	32 ± 11	9 (45%)	24.4 ± 4.3	prospective observational	Evaluation of inert gas rebreathing for determination of cardiac output: influence of age, gender, and body size	__	IGR using the Innocor device provides repeatable measurements of CO and related indices, which are sensitive to the effects of acute physiological maneuvers. Moreover, IGR is a suitable technique for examining chronic influences such as age, gender, and body size on key hemodynamic components of the arterial blood pressure.
Schmid J et al., 2015 * [102]	Bern University Hospital and University of Bern, Switzerland	29	IGR	**Test:** progressive work rate; **Equipment:** cycle ergometer; **Watt range/protocol:** 15 W/2 min; **Cadence:** -; **IGR measurement:** at the end of each step; **Other:** measurements conducted at 540 m and 3454 m altitude.	HF (IHD—ischemic heart disease, *n* = 19; ICD—with an implantable cardioverter defibrillator, *n* = 10)	60.0 ± 8.9	25 (86%)		prospective observational	Influence of water immersion, water gymnastics, and swimming on cardiac output in patients with heart failure	IGR as secondary investigation	Although cardiac index and V˙ O_2_ are lower than in patients with coronary artery disease with preserved left ventricular function and controls, these patients are able to increase cardiac index adequately during water immersion and swimming.

* the extraction of raw data was unattainable.

**Table 3 jcm-12-07154-t003:** Results of the systematic review (with raw data extracted where feasible).

Author and Year	Disease/State of Health	Rest CO (L/min)—Healthy	Rest CO (L/min)—Disease	Peak CO (L/min)—Healthy	Peak CO (L/min)—Disease	Rest CI (L/min/m^2^)—Healthy	Rest CI (L/min/m^2^)—Disease	Peak CI (L/min/m^2^)—Healthy	Peak CI (L/min/m^2^)—Disease	Rest SV (mL/beat)—Healthy	Rest SV (mL/beat)—Disease	Peak SV (mL/beat)—Healthy	Peak SV (mL/beat)—Disease	Rest VO_2_ (mL/kg/min)—Healthy	Rest VO_2_ (mL/kg/min)—Disease	Peak VO_2_ (mL/kg/min)—Healthy	Peak VO_2_ (mL/kg/min)—Disease	Rest VO_2_ (L/min)—Healthy	Rest VO_2_ (L/min)—Disease	Peak VO_2_ (L/min)—Healthy	Peak VO_2_ (L/min)—Disease	Rest ΔC(a-v)O_2_—Healthy	Rest ΔC(a-v)O_2_—Disease	Peak ΔC(a-v)O_2_—Healthy	Peak ΔC(a-v)O_2_—Disease
Koshy A. et al., 2019 [65]	HF		5.32 ± 1.28		10.34 ± 3.14						77.83 ± 23.47		91.43 ± 40.77		3.99 ± 1.17		19.04 ± 5.64		0.33 ± 0.10		1.57 ± 0.42				
Fontana P et al., 2011 [66]	none (healthy and non-smokng)			18.2 ± 3.3 (WT) 17.9 ± 2.6 (GXT)								127 ± 37 (WT) 94 ± 15 (GXT)				45.0 ± 5.3									
Corrieri N et al., 2021 [67]	231 HF patients (HF), 265 healthy volunteers (HV)	5.5 4.6–6.6	3.5 2.8–4.2	13.6 11.2–16.0	6.7 5.2–8.5	3.1 2.6–3.6	1.8 1.6–2.2	7.5 6.7–8.5	3.5 3.0–4.5	67 55–84 (mL)	50 41–63 (mL)	85 71–104 (mL)	65 53–85 (mL)	4.7 4.0–5.5	4.3 3.6–4.7	28.1 23.7–34.2	14.1 11.4–16.8	0.314 0.279–0.376	0.300 0.256–0.350	1.953 1.522–2.579 (mL/min)	1.018 0.800–1.323	5.8 4.9–7.6 (mL/100 mL)	9.1 7.4–10.6 (mL/100 mL)	14.6 10.9–16.3 (mL/100 mL)	15.6 13.8–17.8 (mL/100 mL)
Shelton R et al., 2010 [68]	23 HF patients (HF), 42 patients without HF (no-HF); seven subjects (four CHF patients and three non-CHF patients with similar demographics) were unable to perform an adequate rebreathing manoeuvre and were excluded from further analysis					2.1 (0.6)	1.8 (0.5)	4.2 (0.7)	3.6 (0.8)									0.27 (0.08)	0.27 (0.08)	1.0 (0.14)	1.02 (0.15)	32.4 (7.4) (%)	37.8 (4.9) (%)	63.0 (12.1) (%)	75.4 (10.4) (%)
del Torto A et al., 2019 [59]	none (young, active, and healthy, non-smokers)	$\dot{Q}$ PF 10.4 ± 1.7 (50 W); $\dot{Q}$ IN 12.6 ± 1.0 (50 W)		$\dot{Q}$ PF 22.9 ± 2.5 (250 W); $\dot{Q}$ IN o 20.8 ± 1.4 (250 W)														1.21± 0.04 (50 W)		3.62 ± 0.06 (250 W)		96.8 ± 6.8 (50 W) (mL/L)		174.7 ± 9.3 (250 W) (mL/L)	
Halbirk M et al., 2010 [31]	HF due to ischemic heart disease; GLP-1 infusion group, Placebo group						GLP-1 1.5 ± 0.1; Placebo 1.7 ± 0.2		GLP-1 4.0 ± 0.6; Placebo 3.9 ± 0.4		GLP-1 40 ± 3; Placebo 46 ± 5		GLP-1 74 ± 12; Placebo 72 ± 8		GLP-1 3.3 ± 0.5; Placebo 3.4 ± 0.6		GLP-1 18 ± 2; Placebo 16 ± 2								
Schmidt T et al., 2018 [69]	HF supported by LVAD		3.76 (±0.9)		6.95 (±1.4)		1.9 (±0.4)		3.5 (±0.7)						3.3 (±0.6)		10.9 (±2.6)						7.4 (±1.9) (mL/dL); 44 (±10) (% of oxygen content)		13.2 (±13.1) (mL/dL); 75 (±12) (% of oxygen content)
Okwose N et al., 2017 [70]	none (healthy)	IGR 6.5 ± 2; BR 7.8 ± 1		IGR 18.5 ± 2,6; BR 21.6 ± 3.3		IGR 3.5 ± 1; BR 4.4 ± 1		IGR 9.2 ± 1.8; BR 10.9 ± 1.5		IGR 90 ± 28; BR 104 ± 14 (mL)		IGR 119 ± 30; BR 124 ± 23 (mL)								2.17 ± 0.7				IGR 16.6; BR 15.5 (mL/dL)	
Apostolo A et al., 2018 [71]	HF patients supported with Jarvik 2000 LVADs		Speed 2 3.18 ± 0.76; Speed 4 3.69 ± 0.75		Speed 2 5.91 ± 1.31; Speed 4 6.69 ± 0.99		Speed 2 1.63 ± 0.35; Speed 4 1.89 ± 0.38		Speed 2 3.05 ± 0.61; Speed 4 3.47 ± 0.50								Speed 3 11.7 ± 2.8; Speed 3-4-5 12.5 ± 2.5				Speed 3 948 ± 238; Speed 3-4-5 1014 ± 219 (mL/min)				
Del Torto A et al., 2017 [72]	HF		3.4 2.8–4.2		6.3 5.1–8.0		1.83 1.55–2.17		3.39 2.87–4.21		50 40–61 (mL)		61 51–75 (mL)		4.0 3.4–4.7		13.4 10.9–16.4		0.3 0.25–0.35		0.96 0.78–1.28		8.86 7.21–10.4 (mL/dL)		16.00 14.17–18.00 (mL/dL)
Vignati C et al., 2019 [73]	HF																				1101 (870–1418)-CPET; 1103 (844–1389)-CPET + CO				
Lang C et al., 2007 [60]	HF		3.5 ± 1.1		7.2 ± 2.7										12.6 ± 4.7										
Morosin M et al., 2018 [74]	HF (MF—muscular fatigue *n* = 132; D—dyspnoea *n* = 38)		MF 3.26 ± 0.98; D 3.12 ± 0.92		MF 6.68 ± 2.51; D 6.21 ± 2.55						MF 48.5 ± 15.1; D 46.6 ± 14 (L/min)		MF 64.5 ± 21.2; D 58.3 ± 16.2 (L/min)				MF 15.49 ± 4.77; D 15.35 ± 4.34				MF 1.21 ± 0.43; D 1.14 ± 0.41		MF 9.87 ± 3.2; D 10.1 ± 2.7 (mL/100 mL)		MF 18.2 ± 3.8; D 18.2 ± 3.6 (mL/100 mL)
Cattadori G et al., 2011 [75]	HF (“Pre”—before—and “Post”—after—an 8-week training program)		Pre 3.7 ± 1.1; Post 3.9 ± 1.1		Pre 6.6 ± 2.2; Post 7.3 ± 2.5						Pre 51 ± 16; Post 54.5 ± 16.9 (mL)		Pre 62.6 ± 25.7; Post 70.0 ± 25.5 (mL)				Pre 14.1 ± 3.9; Post 15.1 ± 4.6				Pre 1.1 ± 0.4; Post 1.2 ± 0.4		Pre 9.8 ± 4.4; Post 8.6 ± 2.6 (mL/100 mL)		Pre 17.5 ± 5.1; Post 16.6 ± 4.1 (mL/100 mL)
Reiss N et al., 2018 [76]	HF supported by LVAD		4.1 ± 1.3		7.6 ± 2.4																				
Vignati C et al., 2017 [77]	HF supported by LVAD		Speed 2 3.4 ± 0.9; Speed 4 3.8 ± 1.0		Speed 2 5.3 ± 1.3; Speed 4 35.9 ± 1.4												Speed 2 10.0 ± 1.6; Speed 4 10.8 ± 2.0				Speed 2 788 ± 169; Speed 4 841 ± 152		Speed 2 11.4 ± 4.5; Speed 4 9.9 ± 2.7 (mL/100 mL)		Speed 2 15.7 ± 3.9; Speed 4 15.1 ± 3.6 (mL/100 mL)
Iellamo F et al., 2013 [78]	postinfarction CHF—chronic heart failure (ACT—aerobic continuous training *n* = 8; AIT—aerobic interval training *n* = 8)		before ACT 3.24 ± 0.63; after ACT 3.38 ± 0.73; before AIT 3.43 ± 0.80; after AIT 3.43 ± 0.77								before ACT 58.29 ± 8.53; after ACT 59.14 ± 6.51; before AIT 59.00 ± 10.35; after AIT 59.67 ± 9.04 (mL)						before ACT 18.44 ± 4.29; after ACT 22.53 ± 3.13; before AIT 18.78 ± 4.58; after AIT 23.02 ± 4.28								
Pokan R et al., 2014 [79]	ultraendurance cycling athletes			1 h 16.9 ± 1.5; 6 h 17.2 ± 1.8; 12 h 17.5 ± 1.7; 18 h 17.6 ± 2.1; 24 h 17.7 ± 2.3												LTP1 36 ± 6; LTP2 54 ± 10; max 68 ± 12				LTP1 2757 ± 427; LTP2 4165 ± 552; max 5231 ± 741 (mL/min)					
Accalai E et al., 2021 [61]	HF—heart failure (*n* = 507), HC—healthy controls (*n* = 500)	5.4 ± 1.5	3.3 ± 1	13.3 ± 3.5	6.6 ± 2.6					68 ± 22.1 (mL)	49.5 ± 16.6	85 ± 21.5	63.8 ± 21.4	4.8 ± 1.18	4.06 ± 1.01	29.5 ± 7.7	14.6 ± 4.7	330.7 ± 90.4 (mL/min)	300 ± 83.8 (mL/min)	2026 ± 668 (mL/min)	1113 ± 413 (mL/min)	6.5 ± 1.9 (ml/100 ml)	9.5 ± 3.1 (mL/100 mL)	15.2 ± 2.7 (mL/100 mL)	17.1 ± 3.9 (mL/100 mL)
Fontana p et al., 2010 [80]	recreationally trained male cyclists	6.4 ± 1.2		18.2 ± 2.3 (103s)						91 ± 23 (mL)		123 ± 21 (103s)						0.43 ± 0.08		3.62 ± 0.27 (103s)					
Prochnau D et al., 2012 [81]	HF + CRT						2.02 ± 0.2		3.38 ± 0.23																
Lang C et al., 2009 [82]	HF				(all patients, *n* = 148) 7.3 ± 2.9; (NYHA1, *n* = 12) 10.2 ± 2.8; (NYHA2, *n* = 38) 9.4 ± 2.7; (NYHA3, *n* = 73) 6.7 ± 2.2; (NYHA4, *n* = 23) 4.6 ± 1.5												(all patients, *n* = 148) 12.9 ± 4.5; (NYHA1, *n* = 12) 21.4 ± 6.3; (NYHA2, *n* = 38) 15.7 ± 3.1; (NYHA3, *n* = 73) 11.6 ± 2.1; (NYHA4, *n* = 23) 8.1 ± 1.9								
Agostoni P et al., 2017 [57]	none (healthy)	5.4 ± 1.5		13.2 ± 3.5		3.0 ± 0.8		7.33 ± 1.59				84.5 ± 21.6 (mL)								2025 ± 668 (mL/min)				15.2 ± 2.7 (mL/100mL)	
Jakovljevic D et al., 2011 [83]	HF—heart failure (*n* = 20); LVAD—implanted LVAD (*n* = 18); exLVAD—explanted LVAD (*n* = 16)				HF 9.1 ± 2.1; LVAD 12.4 ± 2.2; exLVAD 14.6 ± 2.9								HF 71.1 ± 14.1; LVAD 87.2 ± 16.4; exLVAD 89.4 ± 15.1				HF 15.8 ± 5.8; LVAD 19.8 ± 4.1; exLVAD 28.2 ± 5.0				HF 1384 ± 375; LVAD 1592 ± 554; exLVAD 2511 ± 468 (mL/min)				
Fontana P et al., 2009 [63]	healthy, asymptomatic, non-smoking, and recreationally active	A1 5.31 ± 0.9; A2 5.41 ± 0.9; B1 5.41 ± 1.1; B2 5.51 ± 1.2		A1 17 ± 3.0; A2 16.6 ± 2.5; B1 16.4 ± 2.2; B2 16.7 ± 2.4																A1 3.18 ± 0.7; A2 3.17 ± 0.72; B1 3.24 ± 0.75; B2 3.23 ± 0.7					
Jakovljevic D et al., 2012 [84]	HF—heart failure		test I 4.0 ± 1.0; test II 3.9 ± 0.9		test I 11.2 ± 5.7; test II 11.3 ± 5.6										test I 3.9 ± 0.9; test II 4.0 ± 0.9		test I 21.7 ± 7.0; test II 23.0 ± 5.8		test I 304 ± 98; test II 320 ± 100 (mL/min)		test I 1792 ± 827; test II 1884 ± 767 (mL/min)				
Okwose N et al., 2019 [51]	healthy	test 1 7.4 ± 1.6; test 2 7.1 ± 1.2		test 1 20.4 ± 2.3; test 2 19.8 ± 2.6						test 1 114 ± 28; test 2 108 ± 15		test 1 135 ± 25; test 2 132 ± 25		test 1 4.4 ± 1.7; test 2 4.7 ± 1.6		test 1 35 ± 1.7; test 2 33.4 ± 0.9		test 1 0.3 ± 0.1; test 2 0.3 ± 0.1		test 1 2.2 ± 0.5; test 2 2.2 ± 0.5					
Vignati C et al., 2021 [85]	patients with PMR (primary mitral regurgitation, *n* = 62) and FMR (funtctional mitral regurgitation, *n* = 53), before (PRE-pMVR) and after (POST-pMVR) pMVR (percutaneous edge-to-edge mitral valve repair)		FMR: PRE-pVMR 3.0 ± 0.8; POST-pMVR 3.4 ± 0.8/PMR: PRE-pVMR 3.1 ± 0.9; POST-pMVR 3.2 ± 0.9		FMR: PRE-pVMR 5.6 ± 1.4; POST-pMVR 6.3 ± 1.5/PMR: PRE-pVMR 6.2 ± 2.4; POST-pMVR 6.7 ± 2.0						FMR: PRE-pVMR 44 ± 13; POST-pMVR 51 ± 14/PMR: PRE-pVMR 43 ± 14; POST-pMVR 45 ± 13 (mL)		FMR: PRE-pVMR 57 ± 19; POST-pMVR 66 ± 20/PMR: PRE-pVMR 62 ± 20; POST-pMVR 69 ± 20 (mL)						FMR: PRE-pVMR 294 ± 64; POST-pMVR 298 ± 56/PMR: PRE-pVMR 269 ± 59; POST-pMVR 276 ± 54 (mL/min)		FMR: PRE-pVMR 932 ± 204; POST-pMVR 968 ± 224/PMR: PRE-pVMR 938 ± 302; POST-pMVR 957 ± 258 (mL/min)		FMR: PRE-pVMR 10.2 ± 2.9; POST-pMVR 9.2 ± 2.3/PMR: PRE-pVMR 9.3 ± 2.7; POST-pMVR 9.2 ± 2.7 (mL/100 mL)		FMR: PRE-pVMR 17.3 ± 4.1; POST-pMVR 15.7 ± 3.7/PMR: PRE-pVMR 15.7 ± 3.8; POST-pMVR 14.9 ± 4.4 (mL/100 mL)
Bentley R et al., 2019 [86]	none (healthy) PRE—before training; POST—after training	PRE 6.3 ± 1.2; POST 6.5 ± 1.0		PRE 21.2 ± 3.5; POST 21.9 ± 3.3						PRE 76 ± 16; POST 79 ± 18						PRE 44.0 ± 6.3; POST 48.1 ± 7.7									
Jakovljevic D et al., 2010 [87]	HF supported by LVAD		optimal speed 5.3 ± 1.7; reduced speed 4.6 ± 1.4		optimal speed 12.2 ± 2.1; reduced speed 8.6 ± 2.5						optimal speed 71.4 ± 16.1; reduced speed 58.2 ± 12.2		optimal speed 88.4 ± 16.4; reduced speed 67.5 ± 14.7		optimal speed 4.3 ± 0.4; reduced speed 4.6 ± 0.3		optimal speed 18.2 ± 4.5; reduced speed 14.1 ± 5.3								
Shen Y et al., 2016 [88]	CHF—chronic heart failure		4.0 ± 1.6		5.8 ± 2.4												14.0 ± 3.9								
Rajani R et al., 2010 [89]	aortic stenosis (AS—asypmtomatic *n* = 28; S—symptomatic *n* = 10)						AS 2.9 ± 0.7; S 2.8 ± 0.8		AS 5.9 ± 1.6; S 4.1 ± 0.9						AS 4.7 ± 2.5; S 5.4 ± 3.2		AS 23.1 ± 5.8; S 16.7 ± 3.8								
Pastormerlo L et al., 2015 [90]	HF (hsT < 20%—hs-Troponin Increase < 20% *n* = 20; hsT > 20%—hs-Troponin Increase > 20% *n* = 10)		hsT < 20% 3.4 ± 1.2; hsT > 20% 3.1 ± 1.2		hsT < 20% 8.3 ± 1.9; hsT > 20% 7.1 ± 1.9																				
Goda A et al., 2009 [58]	HF		4.5 ± 1.6		8.8 ± 2.8										3.8 ± 1.0		15.4 ± 4.4						9.5 ± 4.3		2.1 ± 0.8
Hassan M et al., 2017 [91]	HF						1.75 (1.0–3.4)		3.1 (1.4–6.9)								15.5 (6.5–25)								
Lee W et al., 2011 [92]	precapillary PH (pulmonary hipertension)		PBF: Before therapy: supine position 4.0 ± 1.0, erect position 3.2 ± 0.6. After therapy: supine position 4.7 ± 1.4, erect position 3.9 ± 0.9		PBF: Before therapy: 5.3 ± 1.7. After therapy: 5.7 (4.1–8.1)		2.2 ± 0.7				Before therapy: supine position 53 ± 22, erect position 40 ± 13. After therapy: supine position 58 (44–68), erect position 50 ± 15 (mL)		Before therapy: 48 ± 18. After therapy: 56 ± 21 (mL)				12 ± 4				0.971 ± 0.355				
Bentley R et al., 2018 [93]	none (healthy and non-smoking); divided post-hoc into: LCR—lower cardiac resoinders (*n* = 10); HCR—higher cardiac responder (*n* = 10)	All: 6.2 ± 1.1; LCR: 6.4 ± 1.5; HCR: 6.0 ± 1.0		LCR: 11.0 ± 2.0; HCR: 17.3 ± 2.3						All: 75 ± 15; LCR: 70.0 ± 20.0; HCR: 83.0 ± 11.0								All: 0.34 ± 0.05; LCR: 0.33 ± 0.05; HCR: 0.33 ± 0.05		All: 3.0 ± 0.5; LCR: 2.6 ± 0.4; HCR: 3.4 ± 0.5					

**Table 4 jcm-12-07154-t004:** Indications for CPET-IGR in included articles.

Indication for CPET-IGR	n. of Articles	% of Articles Included
1. HF (heart failure):	25	53%
LVAD pump	6	
2. MR (mitral regurgitation)	1	2%
3. AS (aortic stenosis)	2	4%
4. PPH (precapillary pulmonary hypertension)	1	2%
5. Assessment of healthy individuals:	13	28%
athletes	2	
6. DMT2 (diabetes mellitus t.2)	3	6%
7. COPD (chronic obstructive pulmonary disease	2	4%

**Table 5 jcm-12-07154-t005:** Proposed CPET-IGR report.

Patient information: Name: Age: Gender: Medical Record Number: Date of Test:			
Indication for CPET + IGR: Provide a summary of the patient’s medical condition or reason for conducting the CPET. This may include a suspected cardiovascular or pulmonary disease, exercise intolerance, preoperative evaluation, or any other relevant clinical indication.			
Protocol: Describe the specific CPET protocol employed during the test. Include details such as the type of ergometer (treadmill or cycle ergometer), ramp or incremental protocol, exercise duration, and any specific adjustments made based on the patient’s condition or limitations.			
Medication taken: [List all medications the patient took before the CPET, including the dosage and frequency. Highlight any medications that may affect cardiovascular or respiratory parameters.]			
Hb: [the hemoglobin level at the time of the CPET measured in g/dL or other appropriate units; the level of Hb impacts the a-vO_2_ diff]			
CPET		IGR	
VO_2_ (rest) [L/min]/[mL/kg/min]	VO_2_ (peak) [L/min]/[mL/kg/min]	CO (rest) [L/min]	CO (peak) [L/min]
HR (rest) [bpm]	HR (peak) [bpm]	SV (rest) [mL/beat]	SV (peak) [mL/beat]
RQ (peak)	% of predicted VO_2_ max [%]	a-vO_2_ diff (rest) [mL/100mL]	a-vO_2_ diff (peak) [mL/100mL]
VE/VCO_2_	Resp. Freq.	PBF (rest) [L/min]	PBF (peak) [L/min]
Load [Watts]	SpO_2_ [%]	Shunt (rest) [%]	Shunt (peak) [%]
Graphs: Include a graph that depicts the relationship between CO and VO_2_ max. Plot the CO values on the y-axis and the VO_2_ max values on the *x*-axis. Each data point represents an individual patient’s result. Additionally, you may consider highlighting the patient’s specific data point on the graph for easier interpretation.			
Interpretation: Provide a detailed interpretation of the CPET and non-invasive CO measurement results, considering the patient’s demographics, indication for the test, medications taken, and reference values when available. Discuss any abnormalities or significant findings in the context of the patient’s clinical presentation. Consider integrating the relevant literature or guidelines to support your interpretation.			
Conclusions: Summarize the key findings of the CPET and non-invasive CO measurement, highlighting any notable results or abnormalities. Provide recommendations for further evaluation or management if applicable.			

## Data Availability

Data are contained within the article.

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
