# Peer review of "Non-Invasive Cardiac Output Measurement Using Inert Gas Rebreathing Method during Cardiopulmonary Exercise Testing—A Systematic Review"

_jcm, 2023, doi:10.3390/jcm12227154_

Round 1
Reviewer 1 Report
Comments and Suggestions for Authors
Revision: “Non-invasive Cardiac Output Measurement using Inert Gas Rebreathing method during Cardiopulmonary Exercise Testing – a systematic review.”
This systematic review analyzes inert gas rebreathing during cardiopulmonary exercise testing, reviewing 47 scientific papers (excluding articles concerning the pediatric population, patients with congenital malformations, articles not in English, and reviews). It aims to standardize the procedures for the non-invasive measurement of IGR as a method of calculating cardiac output and proposes a protocol for performing the exercise test and IGR during it. The papers’ selection’s methods are consistent.
There are no systematic reviews in the scientific literature that deal with this topic. This makes the work interesting and original. The text is easy to understand, although many sentences are too long. The references are varied, ranging from the seventies of the last century to 2023, the majority of the references are published in the last fifteen years.
The authors' proposal is to create guidelines on the measurement of cardiac output during the cardiopulmonary exercise test.
The authors recommend as normal values the values reported in Agostoni's work (reference 59), defined as the only work to propose normal IGR values.
As stated by the authors, a considerable limitation is the small sample of patients in the analyzed articles.
Major comments:
- Although it is a systematic review, the manuscript totally lacks data and reference values. The values of Agostoni's article taken into consideration are not even mentioned. Moreover, the authors should add a comprehensive table with all the study included (method used, population included, goal of the study comparison with other methos, results, number of patients) and they should describe the studies or the main results of the studies.
- It should be specified in which of the papers that are analyzed, an alteration of the IGR parameters, compared to the reference values, resulted in mortality or an increased cardiovascular risk.
Minor comments:
- Use shorter periods.
- Abbreviations: after having defined the abbreviation it should be used as such in the text and not in full. Furthermore, all abbreviations must be reported in the specific paragraph. Complete the abbreviations.
- Some typos are worth mentioning:
- line 38, 122, 123: a different character was used
- line 40: final point
- line 115: repetition “al. al.”
- paragraph 1.3: there are too little references in the paragraph
- paragraph 3.1: there aren't table 1 and table 2 that you have indicated in the text
- figure 4: “hypertension”, not “hipertension”
- paragraph 3.4 is redundant because the concept is already expressed among a previous paragraph
- paragraph 3.5: LVAD is for?
- paragraph 3.5: too little references to explain this concept of repeatability
- figure 5: PBF is for?
- Paragraph 5: insert some references
Comments on the Quality of English Language
The authors need to make some minor corrections to make the paper more accessible.
Author Response
Dear Editors,
Thank you for your time and attention to our work. Below you will find detailed responses to each reviewer's comments. We hope that they will be comprehensive and will enable further revision of our systematic review.
In addition, we would like to inform that during the revisions of the text, several duplicate references were identified, which were carefully reviewed and revised so as to avoid repetitions.
Best regards
The Authors
Response to reviewer #1
Q1: This systematic review analyzes inert gas rebreathing during cardiopulmonary exercise testing, reviewing 47 scientific papers (excluding articles concerning the pediatric population, patients with congenital malformations, articles not in English, and reviews). It aims to standardize the procedures for the non-invasive measurement of IGR as a method of calculating cardiac output and proposes a protocol for performing the exercise test and IGR during it. The papers’ selection’s methods are consistent.
There are no systematic reviews in the scientific literature that deal with this topic. This makes the work interesting and original. The text is easy to understand, although many sentences are too long. The references are varied, ranging from the seventies of the last century to 2023, the majority of the references are published in the last fifteen years.
The authors' proposal is to create guidelines on the measurement of cardiac output during the cardiopulmonary exercise test.
The authors recommend as normal values the values reported in Agostoni's work (reference 59), defined as the only work to propose normal IGR values.
As stated by the authors, a considerable limitation is the small sample of patients in the analyzed articles.
A1: Dear Reviewer, thank you for your time and attention to our work. It is a great pleasure to note that you find our work original and interesting. We are also pleased that the purpose of our work has been read correctly.
Major comments:
Q2: - Although it is a systematic review, the manuscript totally lacks data and reference values. The values of Agostoni's article taken into consideration are not even mentioned. Moreover, the authors should add a comprehensive table with all the study included (method used, population included, goal of the study comparison with other methos, results, number of patients) and they should describe the studies or the main results of the studies.
A2: All the mentioned data can be found in the tables included in the supplementary materials.. The reference numbers according to the literature list have been added in a revised version of Table 1 (which has also been split into two as suggested also by the second reviewer to improve readability - new Tables 2 and 3). The reference data on IGR values from Agostoni's paper have been added as a separate table (new Table 1).
Q3: It should be specified in which of the papers that are analyzed, an alteration of the IGR parameters, compared to the reference values, resulted in mortality or an increased cardiovascular risk.
A3: The predictive value of IGR in relation to mortality and/or heart failure progression was added as an entry in the comments column of Table 2. In the systematic review, we did not find enough papers to include this data as a separate column in our description. A relevant paragraph on this fact as a contribution to further research has been added in the discussion section (paragraph 4.1).
Minor comments:
Q4:Use shorter periods.
A4: sentences where we have deemed it necessary have been reworded and shortened accordingly so as not to alter the logic of the argument.
Q5: Abbreviations: after having defined the abbreviation it should be used as such in the text and not in full. Furthermore, all abbreviations must be reported in the specific paragraph. Complete the abbreviations.
A5: the relevant missing abbreviations have been added in the text. A list of the abbreviations used has been added to the body text
Q6: Some typos are worth mentioning:
- line 38, 122, 123: a different character was used
- line 40: final point
- line 115: repetition “al. al.”
- paragraph 1.3: there are too little references in the paragraph
- paragraph 3.1: there aren't table 1 and table 2 that you have indicated in the text
- figure 4: “hypertension”, not “hipertension”
- paragraph 3.4 is redundant because the concept is already expressed among a previous paragraph
- paragraph 3.5: LVAD is for?
- paragraph 3.5: too little references to explain this concept of repeatability
- figure 5: PBF is for?
- Paragraph 5: insert some references
A6: the indicated spelling mistakes have been appropriately corrected, the tables are included in the revised version of the manuscript as supplementary materials, paragraph 3.4 has been appropriately deleted in order to avoid redundancy in the text
Q7: The authors need to make some minor corrections to make the paper more accessible
A7: the quality of the written language is assessed by a native speaker and appropriate corrections are made to the text
Reviewer 2 Report
Comments and Suggestions for Authors
This work addresses a generally well-written literature review in a still explorative methodology in performing CPET.
Even if appreciate the effort in describing the procedures and the tentative of develop a standard protocol for the method, some issues remains.
- especially Table 1 is very complex and hard to read, I suggest dived in two tables: one which includes general data (that is substantially the actual table 2, and now there is a redundance) and one which specifices the test results-
- more informations on the design of singles studies should be reported: are observational, retrospectical, prospective studies? Was the study of the IGR the primary outcome or a secondary investigation? The setting? Inclusion and eslcusion criteria?
- a brief summary of general caratheristics of patients (age, sex) and of the included studies should be included into the text
Even if quality assessment is manual, you should report it in results. also, the references values proposed by Arrigoni sholud be reported, with some analisys of differences between them and values reported in the other studies, when available.
a short report of the number of patients included overall is useful, as if present as the results of CPET where used (survival? indication for surgical interventions?)
all the figures should have a dedicated description
Author Response
Dear Editors,
Thank you for your time and attention to our work. Below you will find detailed responses to each reviewer's comments. We hope that they will be comprehensive and will enable further revision of our systematic review.
In addition, we would like to inform that during the revisions of the text, several duplicate references were identified, which were carefully reviewed and revised so as to avoid repetitions.
Best regards
The Authors
Response to reviewer #2
This work addresses a generally well-written literature review in a still explorative methodology in performing CPET.
Even if appreciate the effort in describing the procedures and the tentative of develop a standard protocol for the method, some issues remains.
Q1:- especially Table 1 is very complex and hard to read, I suggest dived in two tables: one which includes general data (that is substantially the actual table 2, and now there is a redundance) and one which specifices the test results-
A1: Dear Reviewer, thank you for your time and attention to our work. It is a great pleasure to note that you find our review well-written. According to your suggestions, we have divided Table 1 into two tables to avoid redundancy (New Table 2 and Table 3, which specifies the results with raw data). We have also decided to Add New Table 1, containing the reference data reported by Agostoni et al.
Q2:- more informations on the design of singles studies should be reported: are observational, retrospectical, prospective studies? Was the study of the IGR the primary outcome or a secondary investigation? The setting? Inclusion and eslcusion criteria?
A2: Appropriate columns concerning the study design, as well as primary and secondary endpoints of the included studies, have been added to Table 2. In our opinion, the most important part of the setting of the study, are the study protocols, reported in Table 2. Adding inclusion and exclusion criteria would make the Table too hard to read and also taking into consideration different patient populations, we have decided that those criteria are not crucial to report.
Q3: - a brief summary of general caratheristics of patients (age, sex) and of the included studies should be included into the text
A3: A description of the participants in the studies included in the systematic review has been added as a separate paragraph 3.2. In addition, data describing the indications for CPET-IGR in the studies has been added in the text of paragraph 3.4.
Q4: Even if quality assessment is manual, you should report it in results. also, the references values proposed by Arrigoni sholud be reported, with some analisys of differences between them and values reported in the other studies, when available.
A4: The manual quality assessment has been described in the text. The reference values on IGR values from Agostoni's paper have been added as a separate table (new Table 1). Also, we have described this study in the text, more precisely (Paragraph 2.4). To our knowledge, the only study analyzing reference values, along with the data collected from other patients' populations (heart failure in this case), is the article titled “Non-invasive estimation of stroke volume during exercise from oxygen in heart failure patients” by Acalai et al. The data can be found in this article, but they are not the main goal of this study, so we decided not to report those measurements in our systematic review.
Q5: a short report of the number of patients included overall is useful, as if present as the results of CPET where used (survival? indication for surgical interventions?)
A5: The number of patients included overall have been included in the text (Paragraph 3.2).
Q6: all the figures should have a dedicated description
A6: The descriptions have been added to each Table and Figure. All Tables and Figures can be found in Supplementary Material.
Round 2
Reviewer 2 Report
Comments and Suggestions for Authors
no other observations